# Breaking the Gradient Barrier: Unveiling Large Language Models for Strategic Classification

**Xinpeng Lv**[1], **Yunxin Mao**[1], **Haoxuan Li**[2], **Ke Liang**[1], **Jinxuan Yang**[3],
**Wanrong Huang**[1], **Haoang Chi**[1], **Huan Chen**[1], **Long Lan**[1], **Yuanlong Chen**[4],
**Wenjing Yang**[1], **Haotian Wang**[1] *

[1]College of Computer Science and Technology, National University of
Defense Technology, Changsha, China
[2]Center for Data Science, Peking University, Beijing, China
[3]Faculty of Engineering, the University of Sydney, Sydney, Australia
[4]Faculty of Computing, Harbin Institute of Technology, Harbin, China
{lvxinpeng, maoyunxin, wenjing.yang, wanghaotian13}@nudt.edu.cn

## Abstract

Strategic classification (SC) explores how individuals or entities modify their features strategically to achieve favorable classification outcomes. However, existing SC methods, which are largely based on linear models or shallow neural networks, face significant limitations in terms of scalability and capacity when applied to real-world datasets with significantly increasing scale, especially in financial services and the internet sector. In this paper, we investigate how to leverage large language models to design a more scalable and efficient SC framework, especially in the case of growing individuals engaged with decision-making processes. Specifically, we introduce *GLIM*, a gradient-free SC method grounded in in-context learning. During the feed-forward process of self-attention, GLIM implicitly simulates the typical bi-level optimization process of SC, including both the feature manipulation and decision rule optimization. Without fine-tuning the LLMs, our proposed GLIM enjoys the advantage of cost-effective adaptation in dynamic strategic environments. Theoretically, we prove GLIM can support pre-trained LLMs to adapt to a broad range of strategic manipulations. We validate our approach through experiments with a collection of pre-trained LLMs on real-world and synthetic datasets in financial and internet domains, demonstrating that our GLIM exhibits both robustness and efficiency, and offering an effective solution for large-scale SC tasks.

## 1 Introduction

As machine learning (ML) algorithms are increasingly applied in high-stakes decision-making domains such as hiring, lending, and college admissions, the need for rapid and accurate adaptation to dynamic inputs has become crucial. When individuals are provided with information about decision rules, they may strategically manipulate their features to achieve favorable outcomes. Such strategic manipulation undermines the performance of ML models and diminishes their reliability. This phenomenon aligns with Goodhart's Law, which states, "Once a measure becomes a target, it ceases to be a good measure" [64]. When decision rules are made public, individuals may adjust their features in ways that exploit the evaluation criteria.

In response to this issue, strategic classification (SC)[30] has emerged as a growing area of research. SC aims to develop algorithms that improve the accuracy of decision models in environments where

---

*Corresponding author

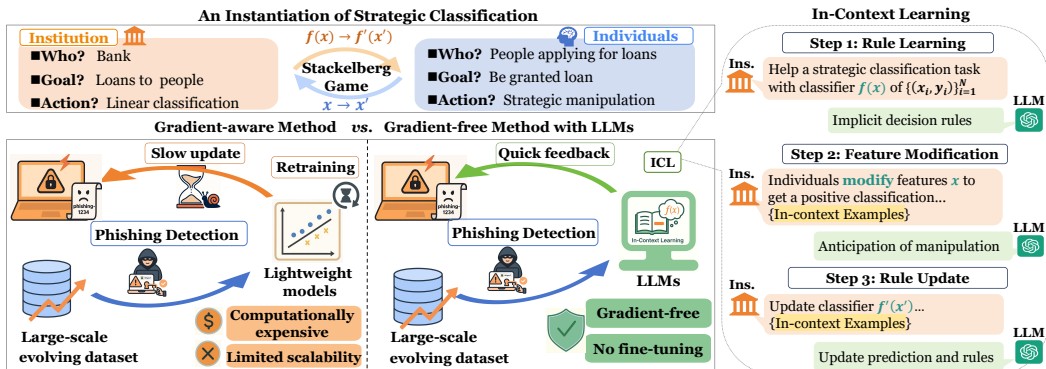

Figure 1: The figure illustrates a strategic classification scenario. Comparison between traditional gradient-based approaches and our gradient-free method using LLMs with ICL for efficient adaptation to Large-scale and evolving data without fine-tuning.

individuals are likely to strategically manipulate their inputs [51, 36, 59]. The SC problem is typically framed as a bi-level optimization [30] following the *Stackelberg game* structure, with the inner and outer optimization objectives referred to as **strategic manipulation** and **decision rule optimization**.

Despite growing theoretical and empirical progress, most existing SC methods, as summarized in Table 1, rely on lightweight models such as linear classifiers or MLPs, and are primarily validated on small-scale datasets (e.g., *Adult* and *Spam*, with fewer than 50,000 samples). However, real-world applications commonly involve significantly larger, dynamically evolving datasets, *often ranging from millions to billions of samples*, rendering existing methods computationally infeasible and inefficient due to their reliance on continuous retraining and explicit gradient computations.

Our work is particularly motivated by data-intensive application domains such as the **internet sector** and **financial services**, where the input distributions shift rapidly due to user interaction or market dynamics, and efficient adaptation to large-scale data is critical. For example, in Figure 1, consider *phishing URL detection*, where attackers continuously modify URLs to evade detection systems. This setting naturally involves large-scale and non-stationary data with adversarial dynamics. Traditional SC approaches often rely on iterative retraining or gradient updates to remain robust, which becomes computationally expensive and infeasible at scale.

In contrast, large language models (LLMs) have demonstrated strong capabilities in modeling high-dimensional and evolving input streams [6, 2], offering a promising foundation for scalable and retraining-free solutions to strategic classification in modern data environments such as fraud detection, credit scoring, spam filtering, and content moderation. However, empowering LLMs with the strategic classification paradigm introduces a unique challenge:

(i) On the one hand, once strategic manipulations lead to changes in individuals' distribution, models for producing decision rules have to be retrained to adapt to the changed distribution [42]. However, when dealing with large-scale data, the cost associated with retraining LLMs becomes prohibitively *expensive and infeasible*.

(ii) On the other hand, without *retraining* the LLMs, it is challenging to model the bi-level optimization of SC, i.e., including the strategic manipulation and the decision rule optimization.

To address these challenges, we propose a novel gradient-free method that leverages in-context learning (ICL) in LLMs to perform strategic classification without updating model parameters. Specifically, we aim to answer the following questions:

> *1. How does ICL simulate strategic manipulations and feature changes in LLMs?*
> *2. How does ICL guide the adjustment of decision rules in LLMs against strategic manipulation?*

Beyond applying ICL to SC tasks, our work theoretically validates the effectiveness of ICL in addressing SC challenges.

**Our primary contributions and findings are summarized as follows:**

Table 1: Comparison of capabilities between existing SC solutions and our proposed method.

| Method | Linear form | Non-linear form | Gradient-free | large-scale data | OOD generalization |
|---|---|---|---|---|---|
| Linear Model [27, 60, 14, 32, 61] | ✓ | ✗ | ✗ | ✗ | ✗ |
| MLP [22, 52, 69] | ✓ | ✓ | ✗ | ✗ | ✗ |
| **GLIM (Ours)** | ✓ | ✓ | ✓ | ✓ | ✓ |

- We theoretically establish, for the first time, how LLMs leveraging in-context learning can implicitly simulate both the strategic manipulation and decision rule optimization stages of the SC bi-level problem, without any fine-tuning.
- Based on this insight, we introduce a **G**radient-free **L**earning **I**n-context **M**ethod *(GLIM)*, that embeds the SC bi-level optimization within pre-trained LLMs, enabling robust and efficient deployment of SC in real-world scenarios.
- We validate our theoretical insights through comprehensive experiments on both synthetic and real-world datasets. The results demonstrate the practical utility and effectiveness of our approach in real-world SC applications.

In Section 2, we introduce the strategic classification task and the in-context learning mechanisms within LLMs. In Section 3, we demonstrate the feasibility of leveraging LLMs for the strategic classification problem and introduce a bi-level implicit gradient descent method for SC. In Section 4, we experimentally validate our theoretical findings and the feasibility of our proposed methods. In Section 5, we review related work on strategic machine learning and large language models.

## 2 Preliminaries

This section introduces the mathematical formulation of strategic classification (SC) and the fundamental concepts of in-context learning (ICL) within LLMs. Throughout our paper, uppercase letters denote random variables (e.g., $X, Y$), while lowercase letters represent their realizations (e.g., $x, y$). Bold symbols (e.g., $\mathbf{x}$ and $\mathbf{X}$) are used for vectors or matrices.

### 2.1 Strategic Classification Task

The SC problem can be formulated as a Stackelberg game [2] involving two players: a **decision maker** (the classifier) and **decision subjects** (the classified individuals) [30, 50].

This setting captures real-world scenarios such as loan approval and college admissions, where institutions publicly announce evaluation criteria, and applicants adapt their features (e.g., test scores, financial statements) towards such criteria. Formally, the decision maker defines a decision rule, e.g., some classifier, $f : \mathbb{R}^d \to \{0, 1\}$, mapping feature vectors to binary outcomes $y \in \{0, 1\}$. Once the rule $f$ is known, individuals may modify their features $\mathbf{x}$ to a new version $\mathbf{x}'$ in hopes of receiving a favorable decision. Such modification incurs a cost, quantified by a cost function $c(\mathbf{x}, \mathbf{x}')$.

In the inner state of this bi-level optimization, each agent aims to maximize their utility, trading off classification benefit with manipulation cost:

**Definition 2.1** (Strategic manipulation in SC tasks). The optimal modified feature vector $\mathbf{x}'$ is determined by:

$$\mathbf{x}' = b(\mathbf{x}) = \arg \max_{x' \in \mathcal{D}} \left[ f(x') - \lambda c(x, x') \right], \tag{1}$$

where $f(x') \in \{0, 1\}$ is the classification result after modification, $c(x, x')$ is the manipulation cost, $\lambda > 0$ is a trade-off parameter, and $\mathcal{D}$ is the feature space. Usually, the cost is modeled as the Mahalanobis Distance $c(\mathbf{x}, \mathbf{x}') = (\mathbf{x}' - \mathbf{x})^\top \mathbf{M}(\mathbf{x}' - \mathbf{x})$, where $\mathbf{M}$ is a Mahalanobis matrix [26, 9].

In the outer stage, the classification rule $f$ is designed to remain robust under such strategic manipulation:

---

[2]In this Stackelberg framework [44], the interaction unfolds in two sequential stages: (i) the decision maker publishes its policy (classification rule $f$), which may be strategic or non-strategic; and (ii) the decision subjects, after recognizing the policy and its associated costs, determine whether to modify their features.

**Definition 2.2** (Decision rule optimization in SC tasks). The decision maker publishes a rule $f^*$ that maximizes accuracy w.r.t modified inputs:

$$f^* \in \arg\max_{f \in \mathcal{F}} \mathbb{E}_{(x,y) \sim \mathcal{D}} \left[ \mathbb{1}\left(f(b(x)) = y\right) \right], \tag{2}$$

where $\mathcal{F}$ refers to the decision function space, and $y$ is the true label.

This objective captures the goal of designing classifiers that remain accurate even when subjects strategically modify their features. In other words, the decision maker aims to anticipate and counteract strategic behavior.

## 2.2 In-Context Learning

In-context learning (ICL) is a paradigm where LLMs perform tasks by conditioning on a small number of labeled examples provided within the input prompt, without requiring any parameter updates. This allows the model to generalize from examples in the input alone, making ICL a flexible and retraining-free strategy for downstream tasks.

**Self-attention.** For a given token $e_j$, its updated embedding through self-attention is [71]:

$$e_j \; \leftarrow \; e_j \; + \; \sum_h \mathbf{P}_h \mathbf{V}_h \operatorname{Softmax}(\mathbf{K}_h^\top \mathbf{q}_{h,j}), \tag{3}$$

where $\mathbf{q}_{h,j}$ is the query vector for head $h$ at position $j$, and $\mathbf{K}_h, \mathbf{V}_h, \mathbf{P}_h$ are learned projection matrices that determine attention scores and output mixing. Bias terms are omitted for clarity.

**ICL as Implicit Gradient Descent.** Recent theoretical progress [2, 1, 73] shows that the forward propagation in LLMs—particularly through linear self-attention layers—can be interpreted as performing *implicit gradient descent (GD)*. Intuitively, this informs that the model learns by simulating an update process internally, even though *no actual change of the parameter weights occurs*.

**Lemma 1** (Forward propagation as implicit gradient descent [1]). *Let $y_\ell^{(n+1)}$ denote the output of the $\ell$-th self-attention layer at token position $(d+1, n+1)$, i.e.,$y_\ell^{(n+1)} = [SA_\ell]_{(d+1),(n+1)}$. Then we have:*

$$y_\ell^{(n+1)} = -\langle x^{(n+1)}, w_\ell^{\mathrm{gd}} \rangle, \tag{4}$$

*where* $\quad w_{\ell+1}^{\mathrm{gd}} = w_\ell^{\mathrm{gd}} - A_\ell \nabla R_{w_\star}(w_\ell^{\mathrm{gd}}), \quad$ *with* $\quad R_{w_\star}(w) := \frac{1}{2n} \sum_{i=1}^n \left( w^\top x_i - w_\star^\top x_i \right)^2.$

This lemma formalizes that ICL can simulate gradient-based learning internally via forward passes, **without explicitly tuning parameters**. Further details on the derivation for ICL are provided in Appendix C, and the proof of this lemma is included in Appendix D.

# 3 A Gradient-free Learning In-context Method for Strategic Classification

## 3.1 LLM-Empowered Strategic Classification

Stemming from [30], strategic classification (SC) can be framed as a bi-level optimization problem (as a Stackelberg framework [44]) where individuals (agents) strategically manipulate their features to receive favorable classification outcomes [3], while the decision maker aims to learn a robust decision rule that anticipates and counteracts such manipulations. To formalize this idea, we recall the bi-level SC problem as stated in section 2.1:

$$\text{Inner Stage (\emph{Strategic manipulation}):} \quad \mathbf{x}' = \arg\max_{\mathbf{x}' \in \mathcal{X}} \left[ f(\mathbf{x}') - \lambda c(\mathbf{x}, \mathbf{x}') \right], \tag{5}$$

$$\text{Outer Stage (\emph{Decision rule optimization}):} \quad f^* = \arg\max_{f \in \mathcal{F}} \mathbb{E}_{(\mathbf{x},y)} \left[ \mathbb{1}\left\{ f(\mathbf{x}') = y \right\} \right]. \tag{6}$$

First, we present two formal definitions to characterize how the two-stage bi-level optimization introduced above is formulated in the language of LLMs:

---

[3]In strategic classification literature, it is commonly assumed that agents are aware of the decision rule. This work adheres to this classical assumption.

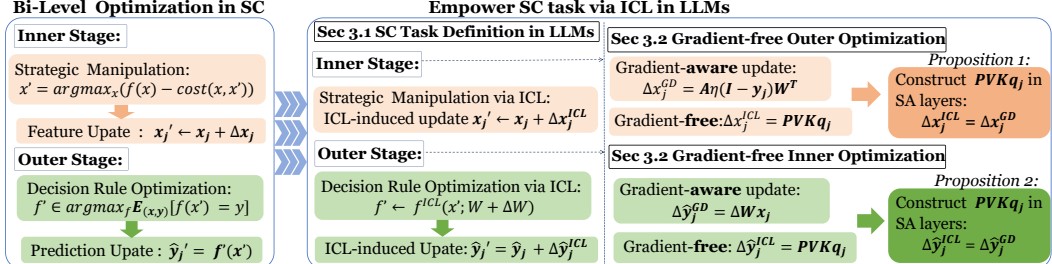

Figure 2: Bi-level optimization in strategic classification is simulated within LLMs, where both inner and outer stage optimizations are realized via ICL.

**LLM-implemented Inner Stage.** With a sequence of labeled prompt examples $\{(\mathbf{x}'_i, y_i)\}_{i=1}^n$, a decision rule $f$ is implicitly defined via attention-based interactions in LLMs. Another feature $\mathbf{x}_j$ is appended as a query token, whose representation evolves through self-attention and yields a manipulated feature $\mathbf{x}'_j$.

**Definition 3.1** (Strategic Manipulation via ICL (***Inner Stage***)). Let $\mathbf{x}_j$ denote an agent's feature and $\{(\mathbf{x}'_i, y_i)\}_{i=1}^n$ be prompt examples. The LLM's forward pass produces a manipulated feature $\mathbf{x}'_j$ as: $\mathbf{x}'_j = \mathbf{x}_j + \Delta\mathbf{x}_j^{\text{ICL}}$, where $\Delta\mathbf{x}_j^{\text{ICL}}$ denotes the feature update implicitly induced by the LLM's self-attention mechanism during ICL.

**LLM-Implemented Outer Stage.** In the *outer stage*, the decision maker aims to optimize the decision rule $f(\cdot; W)$ based on manipulated features $\mathbf{x}'$. In our gradient-free framework, this process is reflected through an ICL-induced shift in predicted scores $\hat{y}_j = f(\mathbf{x}'_j; W)$, effectively capturing the **implicit** optimization of the outer-stage decision rule $f$ and the decision weight $W$.

**Definition 3.2** (Decision Rule Optimization via ICL (***Outer Stage***)). Let $f(\cdot; W)$ be the decision rule implicitly encoded in the LLM. When exposed to manipulated input $\mathbf{x}'$, the classifier's response adapts through in-context prediction, resulting in:

$$\hat{y}' = f^{\text{ICL}}(\mathbf{x}'; W), \tag{7}$$

where $\hat{y}'$ denotes the updated prediction, induced by prompt-driven interactions within the self-attention layers.

Existing SC approaches solve such a bi-level optimization problem through explicit gradient descent [30, 32, 59], i.e., **by tuning the decision models**. However, fine-tuning a large pre-trained model such as LLaMA or DeepSeek incurs prohibitive computational costs. Instead, we propose to implement this two-stage bi-level optimization process of SC task **by leveraging the connection between the ICL and implicit GD, without requiring any parameter updates or fine-tuning.**

### 3.2 Gradient-free Strategic Manipulation via ICL

This subsection provides a theoretical justification for how LLMs equipped with ICL can simulate the strategic manipulation of agents (as the inner stage). Specifically, we show that the feature update $\Delta\mathbf{x}$ obtained from a feed-forward linear self-attention layer **matches** the update derived from traditional gradient descent in strategic classification. For clarity in analysis, we adopt a standard assumption in SC formulations [30, 50, 61]: the decision rule $f(\cdot)$ is assumed to be linear, i.e., $f(\mathbf{x}) = W, \mathbf{x}$ [4].

**Gradient-aware Inner Optimization in Traditional SC.** Conventionally, solving $\Delta x$ in strategic manipulation may be viewed as a gradient-descent step with a learning rate $\eta$ and a loss function $\mathcal{L}_{\text{GD}}$ for Eq. (5):

$$\Delta\mathbf{x} = -\eta\nabla_{\mathbf{x}}\mathcal{L}_{\text{GD}}(\mathbf{x}; W) \quad \Rightarrow \quad \Delta\mathbf{x}_j^{\text{GD}} = A \cdot \eta(1 - y_j)W^\top, \tag{8}$$

where $\mathcal{L}_{\text{GD}}$ is instantiated as a manipulation-aware loss:

$$\mathcal{L}_{\text{GD}}(\mathbf{x}, \mathbf{x}'; W) = \frac{1}{N}\sum_{j=1}^{N}\left[y_j \cdot c(\mathbf{x}_j, \mathbf{x}'_j) + (1 - y_j) \cdot \left(1 - f(\mathbf{x}'_j; W) + \lambda c(\mathbf{x}_j, \mathbf{x}'_j)\right)\right], \tag{9}$$

---

[4]However, our further real-world study using LLMs also verify the superiority of our proposed method in the non-linear regime.

where $A$ is a coefficient matrix that depends on $y_j$ and the manipulation cost function $c$ [5].

**Gradient-free Inner Optimization via ICL.** We now demonstrate that LLMs can implicitly realize the same $\Delta \mathbf{x}$ through forward-only propagation without explicit gradient descent. Consider a linear SA layer [6] applied to token $(\mathbf{x}_j, y_j)$:

$$(\mathbf{x}'_j, y_j) = (\mathbf{x}_j, y_j) + \mathbf{PVK}^\top \mathbf{q}_j, \tag{10}$$

where $\mathbf{q}_j$ is the query vector derived from $\mathbf{x}_j$, and $\mathbf{K}, \mathbf{V}, \mathbf{P}$ are learned key, value, and projection matrices, respectively. Thus, the feature modification during the forward-only propagation, which we termed as *ICL-induced update*, can be written as:

$$\Delta \mathbf{x}_j^{\text{ICL}} = \mathbf{PVK}^\top \mathbf{q}_j. \tag{11}$$

Then we prove that there exists pre-conditioned self-attention weights $\mathbf{P}, \mathbf{V}, \mathbf{K}$, and query vectors $\mathbf{q}_j$ such that $\Delta \mathbf{x}_j^{\text{ICL}} = \Delta \mathbf{x}_j^{\text{GD}}$ (see detailed derivation in Appendix F):

**Proposition 1** (ICL Implements the Gradient-free Strategic Manipulation.). *Let $f(\mathbf{x}; W)$ be a linear classifier. Then, there exists $\mathbf{P}, \mathbf{V}, \mathbf{K}$ such that for any input $\mathbf{x}_j$, the ICL-induced update satisfies:*

$$\Delta \mathbf{x}_j^{ICL} = \Delta \mathbf{x}_j^{GD}, \quad \text{where } \Delta \mathbf{x}_j^{ICL} := \mathbf{PVK}^\top \mathbf{q}_j, \ \ \Delta \mathbf{x}_j^{GD} = A \cdot \eta (1 - y_j) W^\top. \tag{12}$$

*Remark* 1. This proposition [7] informs that LLMs equipped with ICL can simulate the agent-side strategic manipulation by performing implicit GD. This establishes a constructive equivalence between explicit strategic manipulation and attention-driven ICL behavior, thereby grounding ICL as a forward-only approximation of inner-stage optimization in SC.

*Remark* 2 (Linear Derivation.). Following previous protocols [2, 16, 73], our theoretical analysis is performed in the linear regime. However, we note that our proposed method is also **compatible with any non-linear attention and transformer** structures, which have also been extensively empirically validated through our comprehensive experiments (in Appendix I).

### 3.3 Gradient-free Decision Rule Optimization via ICL

This subsection provides a theoretical justification for how LLMs equipped with ICL can simulate the *outer-stage optimization* in strategic classification. Specifically, we demonstrate that the prediction update $\Delta \hat{y}_j$, which reflects a shift in the classifier's decision rule ($f$), can be implicitly implemented via a forward pass in a self-attention layer, without requiring explicit gradient descent or parameter updates.

*Remark* 3. The predicted score is denoted as $\hat{y}_j = f(\mathbf{x}'_j; W) = W\mathbf{x}'_j$, where $W \in \mathbb{R}^d$ is the decision weight vector and $\mathbf{x}'$ is the manipulated feature.

**Gradient-aware Outer Optimization in Traditional SC.** Under standard SC settings, the outer-level decision rule optimization is performed by minimizing a classification loss. For example, using a cross-entropy loss $\mathcal{L}_f$, the update to $W$ via gradient descent is:

$$\Delta W = -\eta \, \nabla_W \, \mathcal{L}_f(W; \mathbf{x}') = \eta \sum_{j=1}^{n} \left( \frac{y_j}{W\mathbf{x}'_j} - \frac{1 - y_j}{1 - W\mathbf{x}'_j} \right) \mathbf{x}'_j, \tag{13}$$

where $\mathcal{L}_f(W; \mathbf{x}')$ is instantiated as:

$$\mathcal{L}_f(W; \mathbf{x}') = -\sum_{j=1}^{n} \left[ y_j \log(W\mathbf{x}'_j) + (1 - y_j) \log(1 - W\mathbf{x}'_j) \right]. \tag{14}$$

Thus, the corresponding shift in prediction output is:

$$\Delta \hat{y}_j^{\text{GD}} = \Delta W \cdot \mathbf{x}'_j. \tag{15}$$

**Gradient-free Outer Optimization via ICL.** We now show that LLMs can reproduce the same prediction update $\Delta \hat{y}_j$ via a forward pass through a self-attention layer. Consider the modified feature

---

[5]See detailed derivation in Appendix E.

[6]The linear self-attention layer is simplified from Eq.(3)

[7]See detailed proof in Appendix F.

vector $\mathbf{x}'_j$, along with the previous predictions $\hat{y}_j$, is embedded into the prompt. The ICL-induced forward update is:

$$(\mathbf{x}'_j, \hat{y}'_j) \leftarrow (\mathbf{x}'_j, \hat{y}_j) + \mathbf{PVK}^\top \mathbf{q}_j, \tag{16}$$

where $\mathbf{q}_j$ is the query vector, and $\mathbf{P}, \mathbf{V}, \mathbf{K}$ are the projection, value, and key matrices.

The update to the prediction output from linear self-attention layers is:

$$\hat{y}'_j = \hat{y}_j + \Delta \hat{y}_j^{\text{ICL}}, \quad \text{where} \quad \Delta \hat{y}_j^{\text{ICL}} := \mathbf{PVK}^\top \mathbf{q}_j. \tag{17}$$

Therefore, we prove that one can construct $\mathbf{P}, \mathbf{V}, \mathbf{K}$ such that:$\Delta \hat{y}_j^{\text{ICL}} = \Delta \hat{y}_j^{\text{GD}}$ [8]:

**Proposition 2** (ICL Implements Gradient-free Decision Rule Update). *Let $f(\mathbf{x}) = \langle W, \mathbf{x} \rangle$ be a linear classifier. Then, there exists a construction of self-attention matrices $\mathbf{K}, \mathbf{V}, \mathbf{P}$ such that for any token $(\mathbf{x}'_j, \hat{y}_j)$, where $\hat{y}_j = f(\mathbf{x}'_j)$, the ICL-induced update satisfies:*

$$\Delta \hat{y}_j^{GD} = \mathbf{PVK}^\top \mathbf{q}_j = \Delta \hat{y}_j^{GD}, \quad where \ \Delta \hat{y}_j^{GD} = \Delta W \cdot \mathbf{x}'_j. \tag{18}$$

*Remark* 4. This proposition[9] confirms that forward-only self-attention dynamics in ICL can simulate gradient-based updates in the outer stage of SC. It establishes a constructive equivalence between explicit decision rule optimization and ICL-driven prediction adaptation.

*Remark* 5 (Unified Simulation of Bi-level Optimization). Together with Proposition 1, this result completes the alignment between ICL and bi-level optimization in SC. ICL enables agent-side manipulation and decision-side rule adjustment, all within a gradient-free, forward-only framework.

### 3.4 Discussion on Policy Transparency

A fundamental characteristic of strategic machine learning is that decision subjects manipulate their input features strategically, understanding the classification rules to achieve more favorable results. This implies that the classification rules should be set transparently to the decision subjects.

Our work is based on theoretical foundations, demonstrating that when LLMs receive a series of in-context prompts, their internal reasoning and output adjustment can be considered an approximate form of "implicit" gradient descent. In other words, although we do not explicitly update the large-scale parameters, LLMs, driven by contextual information such as "*which features to be more sensitive to*" and "*how to define decision boundaries*," adjust their self-attention layers to align with downstream task expectations. Therefore, strategic machine learning based on large language models can also maintain policy transparency. A more detailed discussion is provided in Appendix H.

## 4 Experiment

### 4.1 Setup

**Dataset.** We evaluate our method on six benchmark datasets, comprising five real-world datasets and one synthetic dataset:

- **Large-scale datasets:** *CISFraud* [63], a large-scale transactional dataset provided by IEEE and an international bank for fraud detection. *PhiUSIIL* [55], a phishing URL detection dataset reflecting adversarial evasion scenarios in cybersecurity. *Synthetic* [46], a synthetic dataset generated using the PaySim simulator, which mimics mobile financial transactions and fraud patterns based on real-world data.

- **Small-scale datasets:** *Adult* [4], a census dataset for predicting whether an individual's income. *Spam* [40], a text-based dataset for binary classification of email messages as spam or not. *Credit* [79], a credit scoring dataset used for predicting the risk of credit default in consumer finance scenarios.

**Methods.** We consider two optimal policies a decision maker can use: 1) "strategic policy" means that models consider and handle possible strategic manipulation. 2) "non-strategic policy" means that models in a strategic context, but do not consider strategic manipulation.

---

[8]See detail derivation in Appendix G.
[9]See full derivation in Appendix G.

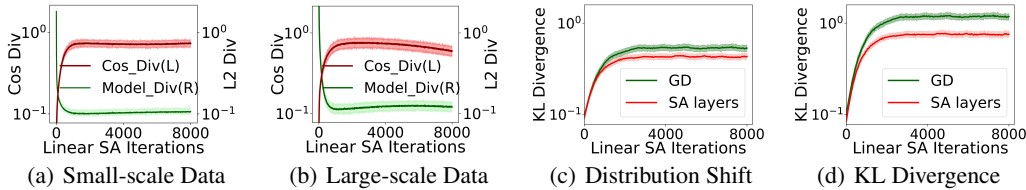

(a) Small-scale Data     (b) Large-scale Data     (c) Distribution Shift     (d) KL Divergence

Figure 3: Comparison of ICL-guided strategic manipulation. (a) and (b) compare ICL and gradient-descent methods across data scales; (c) and (d) evaluate implicit gradient alignment via distribution metrics.

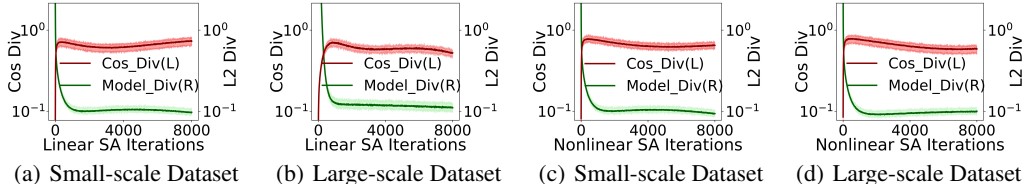

(a) Small-scale Dataset     (b) Large-scale Dataset     (c) Small-scale Dataset     (d) Large-scale Dataset

Figure 4: Comparison of ICL-guided decision rule optimization with Linear and non-linear self-attention layers across dataset scales.

For the baseline method, we employ a linear regression model as a reference classifier, optimizing it through gradient descent. In GLIM, we mainly utilize the pre-trained LLM APIs, e.g., GPT-4o [53], and refine its responses through in-context learning. Each method is subjected to 10-fold cross-validation, and the average results are presented in Table 2. We also conducted experiments on Claude [3], Mixtral [38], DeepSeek [45],Gemini [65], Qwen3 [13], and LLama [49]. Detailed implementation specifics are provided in Appendix J.

## 4.2 Verification on Strategic Manipulation as Implicit Gradient in ICL

To validate the effectiveness of ICL in guiding strategic manipulation through the gradient-free method, we measure both cosine similarity and L2 distance between the feature vectors updated by ICL and those produced by gradient descent. These measurements are conducted under both linear and non-linear settings across different datasets. The results, presented in Figure 3(a) and 3(b), show that the cosine similarity for both methods eventually converges to approximately the same value after some fluctuations, while their L2 distances decrease to nearly zero.

We also compare the mean offset of the feature distribution (distribution shift) and the KL divergence[70] across iterations, as illustrated in Figure 3(c) and 3(d). The close alignment of the two curves confirms that ICL-guided manipulation within the self-attention layers performs comparably to gradient descent. More results are included in Figure 6 of Appendix I.

## 4.3 Verification on Decision Rule Optimization as Implicit Gradient in ICL

To examine how effectively ICL serves as a gradient-free solution for decision rule optimization, we compare the cosine similarity and L2 distance during the optimization process, as depicted in Figure 4 with linear and non-linear attention mechanisms. Across multiple datasets, the two methods exhibit a cosine similarity that gradually rises toward 0.95, while their L2 distances settle at approximately 0.1. These findings suggest that ICL can successfully optimize decision rules via implicit gradients.

Furthermore, we compare how cross-entropy loss evolves in LLMs with GLIM versus the methods via gradient descent for decision optimization. The corresponding results appear in Figure 5(a) and 5(b) with different datasets: a similar loss trend emerges for gradient-free and gradient-aware methods. However, as illustrated in Figure 5(b), when applied to large-scale datasets, the loss reduction attained by LLMs with GLIM surpasses that of existing approaches. These results confirm that it is entirely feasible to employ our proposed method for strategic classification tasks using LLMs.

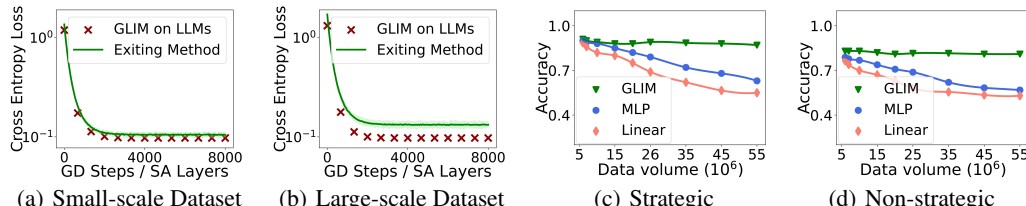

(a) Small-scale Dataset    (b) Large-scale Dataset    (c) Strategic    (d) Non-strategic

Figure 5: (a) and (b): comparison of cross-entropy losses between ICL and gradient-based methods. (c) and (d): comparison of *GLIM* with existing models as the data volume continuously increases.

Table 2: Performance Comparison between GLIM and Existing Methods under Strategic and Non-strategic Settings across Datasets.

| Methods | | Large-scale Dataset | | | Small-scale Dataset | | |
|---|---|---|---|---|---|---|---|
| | | *PhiUSIIL* | *CISFraud* | *Synthetic* | *Credit* | *Adult* | *Spam* |
| **Existing methods (as shown in Table 1)** | | | | | | | |
| *Linear Model* | Strategic | $63.20_{\pm1.02}$ | $63.61_{\pm1.20}$ | $65.50_{\pm2.18}$ | $75.52_{\pm0.60}$ | $77.10_{\pm1.58}$ | $89.67_{\pm0.72}$ |
| | Non-Strategic | $57.39_{\pm0.62}$ | $56.63_{\pm1.08}$ | $60.87_{\pm2.11}$ | $70.73_{\pm0.31}$ | $72.16_{\pm1.62}$ | $87.52_{\pm0.58}$ |
| *MLP* | Strategic | $65.65_{\pm1.14}$ | $65.04_{\pm1.27}$ | $70.90_{\pm2.49}$ | $77.06_{\pm0.41}$ | $78.74_{\pm1.83}$ | $91.05_{\pm0.54}$ |
| | Non-Strategic | $59.25_{\pm0.57}$ | $59.03_{\pm1.06}$ | $65.39_{\pm2.03}$ | $71.50_{\pm0.33}$ | $73.57_{\pm1.55}$ | $89.01_{\pm0.69}$ |
| **GLIM (ours)** | | | | | | | |
| *DeepSeek-V3* | Strategic | $85.10_{\pm0.98}$ | $84.62_{\pm1.09}$ | $85.15_{\pm2.18}$ | $89.33_{\pm0.35}$ | $86.22_{\pm1.34}$ | $94.85_{\pm0.67}$ |
| | Non-Strategic | $78.90_{\pm1.01}$ | $78.74_{\pm1.14}$ | $80.68_{\pm2.12}$ | $\mathbf{81.45}_{\pm0.41}$ | $78.77_{\pm1.33}$ | $89.31_{\pm0.68}$ |
| *GPT-4o* | Strategic | $\mathbf{86.50}_{\pm0.91}$ | $\mathbf{86.89}_{\pm1.08}$ | $\mathbf{86.83}_{\pm2.35}$ | $\mathbf{89.64}_{\pm0.27}$ | $\mathbf{91.35}_{\pm1.29}$ | $\mathbf{95.97}_{\pm0.61}$ |
| | Non-Strategic | $\mathbf{79.14}_{\pm0.94}$ | $\mathbf{80.15}_{\pm1.10}$ | $\mathbf{81.19}_{\pm2.19}$ | $80.96_{\pm0.44}$ | $\mathbf{80.23}_{\pm1.31}$ | $\mathbf{91.28}_{\pm0.65}$ |
| *Claude-3.7* | Strategic | $85.07_{\pm0.95}$ | $84.98_{\pm1.08}$ | $84.50_{\pm2.11}$ | $86.51_{\pm0.31}$ | $88.58_{\pm1.51}$ | $94.50_{\pm0.66}$ |
| | Non-Strategic | $78.40_{\pm0.83}$ | $78.54_{\pm1.17}$ | $78.89_{\pm2.00}$ | $80.39_{\pm0.37}$ | $83.85_{\pm1.50}$ | $89.50_{\pm0.61}$ |

*Note:* 1) We selected linear models and MLPs as representative lightweight approaches from existing methods. 2) The values represent accuracy (%) with standard deviations indicated after the $\pm$ sign. We highlight the best performing results in **bold**. 3) More complete experimental results are included in Tables 3 (in Appendix K).

## 4.4 Analysis on GLIM

Figures 5(c) and 5(d) demonstrate that as data volume increases, the performance of lightweight models becomes less stable, while the proposed **GLIM** method maintains consistent scalability. Table 2 summarizes the overall classification performance across various datasets under both *Non-Strategic* and *Strategic* settings. These results collectively indicate that applying **GLIM** enables large language models to effectively handle strategic classification (SC) tasks, maintaining robustness even when agents engage in strategic manipulations.

Specifically, on the large-scale *PhiUSIIL* dataset under the *Strategic* setting, GPT-4o with **GLIM** achieves an accuracy of 86.50%, showing the model's strong capacity to adapt to strategic inputs. Moreover, on the *Adult* dataset, accuracy increases by 8.36% from the *Non-Strategic* to the *Strategic* setting when equipped with **GLIM**, suggesting that the mechanism not only preserves but also enhances decision robustness under strategic influence. Overall, these findings verify that **GLIM** allows large language models to generalize SC-related reasoning effectively across datasets of different scales and complexities. More experimental results are included in Appendix K.

## 5 Related Work

### 5.1 Strategic Machine Learning

In the realm of strategic classification [30], many studies aim to mitigate strategic manipulations exhibited by individuals interacting with decision models [19, 60, 10, 33, 82, 69]. Building on strategic classification, performative prediction [54, 57, 29, 31, 48, 52] has been proposed to study settings where the deployment of a predictive model influences the distribution of the prediction target. Recently, more studies have explored the role of causal reasoning in strategic machine

learning [50, 9, 34, 72, 21, 8, 78, 76, 74, 75], distinguishing between manipulable and improvable features while accounting for how strategic manipulations may alter underlying qualifications. Other works shift the focus to social welfare [28, 23, 77], aiming to regulate the strategic behavior of agents to maximize social welfare. To avoid disproportionate disadvantage on certain demographic groups, ongoing research also investigates fairness in strategic machine learning [81, 24, 39]. More related work is discussed in Appendix B.

## 5.2 Large Language Model

Recently, large language models (LLMs) [6], with strong in-context learning (ICL capabilities [68, 11], have been applied across a wide range of domains beyond traditional NLP tasks. For example, LLMs have shown significant potential in education [37], medicine [67], and various scientific fields [5]. A recent work has broadened the study of large language models by incorporating them into auction mechanisms [20]. Other studies [7, 47] have explored the use of external tools to enhance the capabilities of LLMs for complex tasks. A series of studies employing linear transformers have demonstrated that forward propagation with ICL in LLMs can internally simulate gradient-based learning mechanisms [2, 73, 16, 17]. Specifically, these models undergo a process analogous to gradient descent by updating the weights in their self-attention layers. To deepen our understanding of ICL, another research direction investigates the problem of learning a function class from in-context examples [25, 12, 41, 80].

## 6 Conclusion

In this study, we demonstrate the feasibility of using large language models (LLMs) to tackle strategic classification problems, which is a pioneering attempt to bridge strategic classification and LLMs via ICL. This is the first to employ LLMs to model and solve the bi-level, game-theoretic optimization structure of SC. Building on this bi-level implicit gradient optimization, our work proposes a gradient-free in-context learning method (*GLIM*) that empowers LLMs to solve strategic classification tasks. It enables a scalable and retraining-free approach to large-scale SC tasks, where classical gradient-based retraining requires excessive computational resources and time. From a broader social science perspective, our work establishes a crucial bridge between large language models and strategic machine learning. Future research will explore the integration of strategic learning within performative prediction frameworks and seek to further enhance policy transparency in LLM-based decision models.

## Acknowledgement

This work is supported in part by the National Natural Science Foundation of China under Grants No. 62276004, 62525213, 62372459, 62376243, 623B2002, and 62302503, the Natural Science Foundation of Heilongjiang Province under Grant No.LH2023C069, the NUDT Youth Independent Innovation Science Fund under Grant No. ZK25-20, the National Key Research and Development Program of China (2024YFE0203700).

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

## A    Clarification on Our Position Towards Gradient-based Methods

We explicitly state that our proposed gradient-free method (GLIM), leveraging large language models and in-context learning, is not intended to criticize or dismiss traditional gradient-based methods widely used in machine learning. Gradient-based optimization has proven extraordinarily effective, well-established, and foundational for machine learning research and applications.

Rather, our work aims to explore and demonstrate an alternative solution path tailored specifically for large-scale strategic classification scenarios, particularly addressing situations where gradient computations and frequent retraining might face practical computational limitations or scalability issues. Our work should be seen as an exploratory contribution, offering additional methodological options to researchers and practitioners, rather than diminishing or replacing the value of gradient-based methods. This work proposes to provide additional methodological options to researchers and practitioners, rather than diminishing or replacing the value of gradient-based methods.

## B    Additional Related Work

There are also some excellent works in the field of strategic machine learning [30] that we did not discuss in Section 5. A previous work [57] proposes lookahead regularization in classification models to anticipate agent behavior during training. To handle inter-user dependencies, incorporating shallow graph neural networks [22] offers a novel pathway for strategic classification. Another work [43] leverages differentiable optimization layers to directly optimize strategic empirical risk in end-to-end systems. Investigating multi-agent strategic settings, [35] proposes classification methods that account for such interdependent effects to improve fairness and robustness. Recently, [62] proposes an optimal stochastic decision rule for strategic classification, demonstrating that introducing randomness into the classifier can effectively reduce classification errors and improve robustness compared to deterministic approaches.

## C    Preliminaries of In-context Learning

Following [58, 73], We review a standard multi-head self-attention ($SA$) layer which updates each element $e_j$ in a set of tokens $\{e_1, \ldots, e_n\}$ according to

$$e_j \leftarrow e_j + SA(j, \{e_1, \ldots, e_n\})$$
$$= e_j + \sum_h \mathbf{P}_h \, \text{softmax} \left( \frac{\mathbf{K}_h^T \mathbf{q}_{h,j}}{\sqrt{d_k}} \right) \mathbf{V}_h, \tag{19}$$

where $\mathbf{P}_h, \mathbf{V}_h, \mathbf{K}_h$ are the projection, value, and key matrices respectively, $d_k$ is the dimension of the key vector and $\mathbf{q}_{h,j}$ is the query vector, all for the $h$-th head.

The columns of the value matrix $\mathbf{V}_h = [\mathbf{v}_{h,1}, \ldots, \mathbf{v}_{h,N}]$ consist of vectors $\mathbf{v}_{h,i} = \mathbf{W}_{\mathbf{V}h} \cdot e_i$, where we introduce $\mathbf{W}_{\mathbf{V}h}$ as the parameter matrix of $\mathbf{V}_h$. Similarly, $\mathbf{k}_{h,1} = \mathbf{W}_{\mathbf{K}h} \cdot e_i$ for the key matrix $\mathbf{K}_h = [\mathbf{k}_{h,1}, \ldots, \mathbf{k}_{h,N}]$ and $\mathbf{q}_{h,j} = \mathbf{W}_{\mathbf{Q}h} \cdot e_j$ for the query vector $\mathbf{q}_h$. These parameters, $\mathbf{P}_h, \mathbf{W}_{\mathbf{V}h}, \mathbf{W}_{\mathbf{K}h}$, and $\mathbf{W}_{\mathbf{Q}h}$ of an $SA$ layer, consist of all projection matrices. The self-attention layer described above corresponds to the one used in standard LLMs and ICL is leveraged to bootstrap the update of these parameter matrices.

During the forward propagation in self-attention layers, ICL aims to leverage the contextual examples $\{(x_i, y_i)\}_{i=1}^n$, embedded into tokens $\{e_i\}_{i=1}^n$ to predict the response for the new query token $e_{n+1}$. Specifically, the model observes an *in-context prompt* composed of $n$ pairs examples and then produces a hypothesis $\hat{y}_{n+1}$ for $e_{n+1}$ Mathematically, one can view the ICL process as inducing a temporary "in-context" mapping $F_{ICL}$ (parametrized by language models) such that:

$$\hat{y}_{n+1} = F_{ICL}(e_{n+1}|e_1, \cdots, e_n). \tag{20}$$

Because this mapping is never explicitly "trained" in the traditional sense (i.e., by gradient descent on the model parameters), the objective of ICL is to minimize the prediction error on the new token using only the in-context examples as guidance. Concretely, the *loss function* for the ICL forward propagation can often be written as:

$$\mathcal{L}_{ICL} = \hat{f}_l(\hat{y}_{n+1}, y_{n+1}), \tag{21}$$

where $\hat{f}_l$ is a task-specific measure of prediction error, e.g., mean-squared error or cross-entropy loss.

## D    Implicit Gradient Descent in Self-Attention Layers

Our Lemma 1 indicates that, under ICL guidance, the token update process within the self-attention layer can be viewed as an implicit gradient optimization process [2].

First, we highlight the dependency on the tokens $e_i$ of the linear self-attention operation

$$
\begin{aligned}
e_j \leftarrow e_j + \mathrm{SA}(\{e_1, \ldots, e_N\}) &= e_j + \sum_h P_h V_h K_h^T q_{h,j} \\
&= e_j + \sum_h P_h \sum_i v_{h,i} \otimes k_{h,i} q_{h,j} \\
&= e_j + \sum_h P_h W_{h,V} \sum_i e_{h,i} \otimes e_{h,i} W_{h,K}^T W_{h,Q} e_j
\end{aligned}
\tag{22}
$$

with $\otimes$ the outer product between two vectors. With this, we can now easily draw connections to one step of gradient descent on $L(W) = \frac{1}{2N} \sum_{i=1}^{N} \|W x_i - y_i\|^2$ with learning rate $\eta$ which yields weight change

$$
\Delta W = -\eta \nabla_W \mathcal{L}(W) = -\frac{\eta}{N} \sum_{i=1}^{N} (W x_i - y_i) x_i^T.
\tag{23}
$$

We provide the weight matrices in block form: $W_K = W_Q = \begin{pmatrix} I_x & 0 \\ 0 & 0 \end{pmatrix}$ with $I_x$ and $I_y$ the identity matrices of size $N_x$ and $N_y$ respectively. Furthermore, we set $W_V = \begin{pmatrix} 0 & 0 \\ W_0 & -I_y \end{pmatrix}$ with the weight matrix $W_0 \in \mathbb{R}^{N_y \times N_x}$ of the linear model we wish to train and $P = \frac{\eta}{N} I$ with identity matrix of size $N_x + N_y$. With this simple construction, we obtain the following dynamics

$$
\begin{aligned}
\begin{pmatrix} x_j \\ y_j \end{pmatrix} &\leftarrow \begin{pmatrix} x_j \\ y_j \end{pmatrix} + \frac{\eta}{N} I \sum_{i=1}^{N} \left( \begin{pmatrix} 0 & 0 \\ W_0 & -I_y \end{pmatrix} \begin{pmatrix} x_i \\ y_i \end{pmatrix} \right) \otimes \left( \begin{pmatrix} I_x & 0 \\ 0 & 0 \end{pmatrix} \begin{pmatrix} x_i \\ y_i \end{pmatrix} \right) \begin{pmatrix} I_x & 0 \\ 0 & 0 \end{pmatrix} \begin{pmatrix} x_j \\ y_j \end{pmatrix} \\
&= \begin{pmatrix} x_j \\ y_j \end{pmatrix} + \frac{\eta}{N} I \sum_{i=1}^{N} \begin{pmatrix} 0 \\ W_0 x_i - y_i \end{pmatrix} \otimes \begin{pmatrix} x_i \\ 0 \end{pmatrix} \begin{pmatrix} x_j \\ 0 \end{pmatrix} \\
&= \begin{pmatrix} x_j \\ y_j \end{pmatrix} + \begin{pmatrix} 0 \\ -\Delta W x_j \end{pmatrix},
\end{aligned}
\tag{24}
$$

for every token $e_j = (x_j, y_j)$ including the query token $e_{N+1} = e_{\text{test}} = (x_{\text{test}}, -W_0 x_{\text{test}})$ which will give us the desired result.

## E    Derivation of Strategic Manipulation via Utility-aligned Loss

In strategic classification, agents modify their features $\mathbf{x}'$ to maximize the utility function:

$$
U(f(\mathbf{x}'), \mathbf{x}') = f(\mathbf{x}') - \lambda c(\mathbf{x}, \mathbf{x}')
\tag{25}
$$

where $f(\mathbf{x}') = \mathbf{W}\mathbf{x}'$ is a linear classifier, and $c(\mathbf{x}, \mathbf{x}') = (\mathbf{x}' - \mathbf{x})^\top \mathbf{M}(\mathbf{x}' - \mathbf{x})$ represents the manipulation cost with $\mathbf{M} \succ 0$.

Given the *sample-wise* loss function:

$$
\mathcal{L}_{\mathrm{GD}}(\mathbf{x}_i') = y_i c(\mathbf{x}_i, \mathbf{x}_i') + (1 - y_i) \left[ 1 - f(\mathbf{x}_i') + \lambda c(\mathbf{x}_i, \mathbf{x}_i') \right]
\tag{26}
$$

The gradient with respect to manipulated features is:

$$
\nabla_{\mathbf{x}_i'} \mathcal{L}_{\mathrm{GD}} = y_i \cdot 2\mathbf{M}(\mathbf{x}_i' - \mathbf{x}_i) + (1 - y_i) \left[ -\mathbf{W}^\top + 2\lambda \mathbf{M}(\mathbf{x}_i' - \mathbf{x}_i) \right]
\tag{27}
$$

Let $\Delta\mathbf{x}_i = \mathbf{x}'_i - \mathbf{x}_i$ denote the feature modification. The gradient descent update with learning rate $\eta$ becomes:

$$\Delta\mathbf{x}_i = -\eta\nabla_{\mathbf{x}'_i}\mathcal{L}_{\text{GD}} \tag{28}$$

**Case 1:** $y_i = 1$ **(Positive Class)**

$$\nabla_{\mathbf{x}'_i}\mathcal{L}_{\text{GD}} = 2\mathbf{M}\Delta\mathbf{x}_i, \tag{29}$$

$$\Delta\mathbf{x}_i = -\eta \cdot 2\mathbf{M}\Delta\mathbf{x}_i, \tag{30}$$

$$(\mathbf{I} + 2\eta\mathbf{M})\Delta\mathbf{x}_i = 0 \implies \Delta\mathbf{x}_i = \mathbf{0}. \tag{31}$$

*Interpretation:* No incentive for manipulation when already classified positively.

**Case 2:** $y_i = 0$ **(Negative Class)**

$$\nabla_{\mathbf{x}'_i}\mathcal{L}_{\text{GD}} = -\mathbf{W}^\top + 2\lambda\mathbf{M}\Delta\mathbf{x}_i, \tag{32}$$

$$\Delta\mathbf{x}_i = \eta\mathbf{W}^\top - 2\eta\lambda\mathbf{M}\Delta\mathbf{x}_i, \tag{33}$$

$$(\mathbf{I} + 2\eta\lambda\mathbf{M})\Delta\mathbf{x}_i = \eta\mathbf{W}^\top. \tag{34}$$

Using the eigendecomposition $\mathbf{M} = \mathbf{Q}\mathbf{\Lambda}\mathbf{Q}^\top$:

$$\Delta\mathbf{x}_i = \eta\mathbf{Q}(\mathbf{I} + 2\eta\lambda\mathbf{\Lambda})^{-1}\mathbf{Q}^\top\mathbf{W}^\top. \tag{35}$$

Define the *adaptation matrix*:

$$\mathbf{A} = (\mathbf{I} + 2\eta\lambda\mathbf{M})^{-1}. \tag{36}$$

yielding:

$$\Delta\mathbf{x}_i = \eta\mathbf{A}\mathbf{W}^\top. \tag{37}$$

Combining both cases:

$$\Delta\mathbf{x}_i = \eta(1 - y_i)\mathbf{A}\mathbf{W}^\top. \tag{38}$$

- $\mathbf{W} \in \mathbb{R}^{1\times d} \implies .\mathbf{W}^\top \in \mathbb{R}^{d\times 1}$.
- $\mathbf{M}, \mathbf{A} \in \mathbb{R}^{d\times d}$.
- $\Delta\mathbf{x}_i \in \mathbb{R}^{d\times 1}$ (dimensionally consistent).

This shows that minimizing the loss function $\mathcal{L}_{\text{GD}}(\mathbf{x};\mathbf{x}')$ results in the same manipulation direction as maximizing the utility function $U(f(\mathbf{x}'),\mathbf{x}')$.

*Remark* 6 (Generality of Cost Function Forms). While our derivation assumes the manipulation cost $c(x, x') = (x' - x)^\top M(x' - x)$ based on the Mahalanobis distance for analytical tractability, our results can be extended to a broader class of distance-based cost functions. In fact, many commonly used cost measures in strategic classification, such as $L_p$ norms, graph distances, and general norms or seminorms, also satisfy the triangle inequality and support similar gradient-based manipulation dynamics [51]. As long as the cost function is differentiable and convex, the gradient-based feature update remains well-defined, and our analysis of in-context manipulation behavior and utility-aligned loss minimization holds under these alternative formulations.

## F  The Self-attention Layer Projection Matrix Constructed for Strategic Manipulation

To demonstrate that the implicit gradient update via in-context learning (ICL) matches the explicit gradient descent step:

$$\Delta\mathbf{x}_j^{\text{GD}} = A \cdot \eta(1 - y_j)W^\top, \tag{39}$$

we provide a constructive setup of the self-attention matrices as follows.

## F.1 Matrix Construction

We define the key, query, and value matrices in block form:

$$W_K = W_Q = \begin{pmatrix} I_x & 0 \\ 0 & 0 \end{pmatrix}, \quad W_V = \begin{pmatrix} 0 & 0 \\ \eta(1 - y_j)W^\top & 0 \end{pmatrix}, \tag{40}$$

where $I_x \in \mathbb{R}^{d \times d}$ is the identity matrix for feature tokens. The projection matrix is defined as:

$$P = \frac{1}{N} \begin{pmatrix} A & 0 \\ 0 & 0 \end{pmatrix}, \tag{41}$$

with $A$ being the manipulation cost-adjusted coefficient matrix from Equation (39), and $N$ is the number of context tokens.

## F.2 Update Dynamics

For a query token $(\mathbf{x}_j, y_j)$, the query vector is formed as:

$$\mathbf{q}_j = W_Q \begin{pmatrix} \mathbf{x}_j \\ y_j \end{pmatrix} = \begin{pmatrix} \mathbf{x}_j \\ 0 \end{pmatrix}. \tag{42}$$

The full attention-based update becomes:

$$\Delta \mathbf{x}_j^{\text{ICL}} = P \cdot VK^\top \mathbf{q}_j = \frac{1}{N} \begin{pmatrix} A & 0 \\ 0 & 0 \end{pmatrix} \sum_{i=1}^{N} W_V e_i \cdot (W_K e_i)^\top \mathbf{q}_j. \tag{43}$$

Unfolding the matrix multiplication:

$$\Delta \mathbf{x}_j^{\text{ICL}} = \frac{A\eta}{N} \sum_{i=1}^{N} (1 - y_i) W^\top \langle \mathbf{x}_i, \mathbf{x}_j \rangle. \tag{44}$$

Assuming homogeneous label groups (i.e., all $y_i = y_j$), we obtain:

$$\Delta \mathbf{x}_j^{\text{ICL}} = A \cdot \eta(1 - y_j) W^\top, \tag{45}$$

which exactly matches the explicit gradient update $\Delta \mathbf{x}_j^{\text{GD}}$.

## F.3 Consistency Verification

The complete feature update under ICL is then:

$$\mathbf{x}_j' = \mathbf{x}_j + \Delta \mathbf{x}_j^{\text{ICL}} = \mathbf{x}_j + A \cdot \eta(1 - y_j) W^\top, \tag{46}$$

confirming equivalence to the explicit manipulation update in Equation (39). The block structure of $W_V$ and $P$ ensures that the label component $y_j$ remains unchanged during the forward pass.

This construction avoids explicit computation of inverse matrices at runtime by directly encoding the gradient dynamics into self-attention weight matrices. The resulting ICL process reflects a forward-only approximation of agent-side strategic behavior within the Transformer framework, and demonstrates the capacity of attention mechanisms to simulate first-order optimization steps relevant to strategic classification.

*Remark* 7 (Connection to Proposition 1). This derivation provides the explicit configuration of matrices $P, V, K$, and query vector $\mathbf{q}_j$, as required in Proposition 1. It confirms that, under linear attention with properly constructed weight matrices, the ICL-induced update $\Delta \mathbf{x}_j^{\text{ICL}}$ exactly matches the explicit gradient response $\Delta \mathbf{x}_j^{\text{GD}}$, thereby validating the proposition with a constructive proof.

## F.4 Extension Beyond the Homogeneous Assumption

The homogeneous label assumption is a theoretical simplification [18] used to illustrate the underlying mechanism of the attention-based update. It enables us to transparently demonstrate how the attention-induced update can precisely align with the gradient descent direction under idealized conditions.

**Derivation Beyond the Assumption.** In practical heterogeneous contexts, the update becomes a weighted aggregation of local update directions:

$$\Delta \mathbf{x}_j^{\text{ICL}} = \frac{A\eta}{N} \sum_{i=1}^{N} (1 - y_i) W^\top \langle \mathbf{x}_i, \mathbf{x}_j \rangle, \tag{47}$$

where $\langle \mathbf{x}_i, \mathbf{x}_j \rangle$ represents the similarity between the query and a context example.

More generally, the attention mechanism implicitly performs a weighted combination of local gradients:

$$\Delta \mathbf{x}_j^{\text{ICL}} = \sum_{i=1}^{N} \alpha_{j,i} \cdot g(\mathbf{x}_i, y_i), \tag{48}$$

where $\alpha_{j,i}$ are the attention weights and $g(\mathbf{x}_i, y_i)$ denotes the local update direction associated with each context sample.

When the context samples are independently drawn and sufficiently representative of the local data distribution around $\mathbf{x}_j$, statistical learning theory [56] ensures that:

$$\lim_{N \to \infty} \Delta \mathbf{x}_j^{\text{ICL}} \approx \mathbb{E}_{(\mathbf{x}_i, y_i) \sim \mathcal{P}_{\text{local}}} \left[ g(\mathbf{x}_i, y_i) \right]. \tag{49}$$

The approximation error can then be bounded as:

$$\epsilon_j = \left\| \Delta \mathbf{x}_j^{\text{ICL}} - \nabla_{\mathbf{x}} \mathcal{L}(\mathbf{x}_j) \right\|, \tag{50}$$

and its expected value satisfies:

$$\mathbb{E}[\epsilon_j] \leq C \cdot \sqrt{\frac{1}{N}} + \delta, \quad \text{with} \quad \delta < L \cdot \mathcal{W}(\mathcal{P}_{\text{local}}, \mathcal{P}_{\text{global}}), \tag{51}$$

where $C \cdot \sqrt{\frac{1}{N}}$ corresponds to the *sampling error*, and $\delta$ captures the *distribution shift*. Specifically, $C$ is a constant depending on the gradient variance, $L$ is the Lipschitz constant of $\mathcal{L}$, and $\mathcal{W}$ denotes the Wasserstein distance between local and global data distributions.

# G   The Self-attention Layer Projection Matrix Constructed for Decision Rule Optimization

We provide a constructive proof of Proposition 2, showing that a single-layer linear self-attention mechanism can simulate the gradient-based update to predictions in the outer-level optimization of strategic classification. To explicitly align the self-attention mechanism with the gradient update $\Delta \hat{y}_j^{\text{GD}}$, we construct the weight matrices as follows:

## G.1   Key and Query Matrices

We construct the key and query matrices to focus exclusively on feature dimensions while ignoring the prediction values:

$$W_K = W_Q = \begin{pmatrix} I_d & 0 \\ 0 & 0 \end{pmatrix} \in \mathbb{R}^{(d+1) \times (d+1)}. \tag{52}$$

Thus, the query vector becomes:

$$\mathbf{q}_j = W_Q \begin{pmatrix} \mathbf{x}_j' \\ \hat{y}_j \end{pmatrix} = \begin{pmatrix} \mathbf{x}_j' \\ 0 \end{pmatrix}, \quad \mathbf{K}^\top = W_K^\top = \begin{pmatrix} I_d & 0 \\ 0 & 0 \end{pmatrix}. \tag{53}$$

## G.2   Value Matrix

The value matrix encodes token-wise gradient terms derived from the cross-entropy loss. For each context token $i$, define the gradient term:

$$\delta_i := \eta \left( \frac{y_i}{W \mathbf{x}_i'} - \frac{1 - y_i}{1 - W \mathbf{x}_i'} \right) \mathbf{x}_i'. \tag{54}$$

Then define the token-specific value matrix:

$$W_V^{(i)} = \begin{pmatrix} 0 & 0 \\ \delta_i & 0 \end{pmatrix} \in \mathbb{R}^{(d+1)\times(d+1)}. \tag{55}$$

Summing over all context tokens gives the full value matrix:

$$\mathbf{V} = \sum_{i=1}^{N} W_V^{(i)} = \begin{pmatrix} 0 & 0 \\ \sum_{i=1}^{N} \delta_i & 0 \end{pmatrix}. \tag{56}$$

### G.3 Projection Matrix

The projection matrix isolates the predicted score dimension (i.e., the final output of the classifier) from the token embedding:

$$\mathbf{P} = \begin{pmatrix} 0 & 0 \\ 0 & 1 \end{pmatrix} \in \mathbb{R}^{(d+1)\times(d+1)}. \tag{57}$$

This ensures that the output update affects only the prediction dimension.

We now compute the full attention-based update through matrix products:

$$\mathbf{VK}^\top = \begin{pmatrix} 0 & 0 \\ \sum_{i=1}^{N} \delta_i I_d & 0 \end{pmatrix},$$

$$\mathbf{VK}^\top \mathbf{q}_j = \begin{pmatrix} 0 \\ \sum_{i=1}^{N} \delta_i^\top \mathbf{x}_j' \end{pmatrix},$$

$$\mathbf{PVK}^\top \mathbf{q}_j = \begin{pmatrix} 0 \\ \eta \sum_{i=1}^{N} \left( \frac{y_i}{W\mathbf{x}_i'} - \frac{1-y_i}{1-W\mathbf{x}_i'} \right) \langle \mathbf{x}_i', \mathbf{x}_j' \rangle \end{pmatrix}.$$

Thus, the second (prediction) component is:

$$\Delta \hat{y}_j^{\text{ICL}} = \eta \sum_{i=1}^{N} \left( \frac{y_i}{W\mathbf{x}_i'} - \frac{1-y_i}{1-W\mathbf{x}_i'} \right) \langle \mathbf{x}_i', \mathbf{x}_j' \rangle = \Delta \hat{y}_j^{\text{GD}}. \tag{58}$$

This construction provides a token-wise simulation of gradient descent over prediction scores using a single self-attention pass, without explicitly modifying $W$. It assumes access to all relevant features $\mathbf{x}_i'$ and uses attention as a proxy for computing interactions $\langle \mathbf{x}_i', \mathbf{x}_j' \rangle$.

*Remark* 8 (Connection to Proposition 2). This derivation offers a complete, constructive realization of the condition in Proposition 2. It demonstrates that the change in predicted score from gradient descent, $\Delta \hat{y}_j^{\text{GD}}$, can be exactly matched by a self-attention layer via $\mathbf{PVK}^\top \mathbf{q}_j$, thereby validating the proposition through an attention-driven forward computation.

## H  Discuss on Policy Transparency in Strategic Machine Learning

Policy transparency is a fundamental concern in strategic machine learning (SC), where individuals strategically manipulate their input features based on their understanding of the classification rules to secure favorable outcomes. Ensuring that classification policies are transparent allows individuals to make informed decisions about how to adjust their features legitimately, thereby maintaining fairness and accountability in the decision-making process.

**Transparency in Traditional SC Models.** Traditional SC models, such as linear classifiers and shallow multilayer perceptrons (MLPs), are preferred in strategic settings due to their inherent interpretability. These lightweight models allow decision-makers to clearly communicate the criteria used for classification, enabling individuals to understand which features are most influential and how they can adjust their inputs accordingly. This transparency not only fosters trust but also helps in mitigating adversarial manipulations by making the decision boundaries explicit and understandable.

**Challenges with LLM-based SC Models.** In contrast, large language models (LLMs) introduce significant challenges to policy transparency. LLMs are characterized by their vast number of parameters and complex architectures, including multiple layers of self-attention mechanisms. This

complexity renders their internal decision-making processes less transparent, making it difficult for users to discern how specific input features influence the final classification outcome. The black-box nature of LLMs can therefore obscure the classification rules, increasing the risk of individuals exploiting hidden patterns or ambiguities to manipulate their features strategically.

**Our Theoretical Contribution: ICL as Implicit Gradient Descent.** Our work addresses this transparency challenge by theoretically demonstrating that *in-context learning* (ICL) within LLMs can be approximated as an implicit gradient descent optimization process. Specifically, we have proven that the iterative adjustments made by LLMs during ICL resemble the steps taken in traditional gradient descent algorithms used in transparent SC models. This approximation provides a conceptual framework for understanding how LLMs adapt to strategic manipulations, offering a semblance of interpretability despite their complex architectures.

**Enhancing Transparency through Attention Visualization.** Building on our theoretical findings, we propose leveraging the self-attention mechanisms inherent in LLMs to enhance policy transparency. By visualizing attention weights, stakeholders can gain insights into which input features the model emphasizes during classification. This visualization acts as a proxy for understanding the decision-making process, allowing users to see how different features contribute to the final classification outcome. Consequently, even though the overall model remains complex, the attention patterns provide a tangible means of interpreting the classification rules.

**Implications and Future Directions.** Our approach offers a pathway to reconcile the powerful modeling capabilities of LLMs with the need for policy transparency in SC tasks. By framing ICL as an implicit gradient descent process and utilizing attention visualizations, we provide a method to interpret and audit LLM-based classification rules effectively. Future research could explore more sophisticated visualization techniques and formalize the interpretability guarantees provided by attention mechanisms. Additionally, developing strategies to balance transparency with the prevention of strategic manipulations remains an important avenue for ensuring both fairness and robustness in LLM-based SC systems.

In conclusion, while LLMs present inherent challenges to policy transparency in strategic classification, our theoretical framework and interpretative techniques offer viable solutions. By understanding ICL as an implicit optimization process and utilizing attention visualizations, we enhance the transparency of LLM-based decision models, ensuring that classification policies remain both effective and comprehensible to users.

# I  Extension to Non-linear Attention Mechanisms

Our theoretical analysis in Section 3 adopts a *linear self-attention* formulation for clarity and analytical traceability. However, modern large language models (LLMs) such as GPT, LLaMA, and DeepSeek operate with a *non-linear multi-head attention mechanism* that uses the Softmax function. In this section, we demonstrate that despite the structural differences, our framework remains applicable in practice and can be naturally extended to non-linear attention.

## I.1  Non-linear Attention in Transformers

In the standard Transformer architecture [71], the update of a token $e_j$ via multi-head self-attention (omitting biases) is:

$$e_j \; \leftarrow \; e_j + \sum_h \mathbf{P}_h \mathbf{V}_h \, \text{Softmax}(\mathbf{K}_h^\top \mathbf{q}_{h,j}), \tag{59}$$

where:

- $\mathbf{q}_{h,j}$ is the query vector of head $h$ at position $j$;
- $\mathbf{K}_h, \mathbf{V}_h$ are the key and value matrices from the prompt;
- $\text{Softmax}(\cdot)$ produces a probability distribution over prompt positions;
- $\mathbf{P}_h$ is the output projection matrix for head $h$.

This process computes a content-dependent weighted average over value vectors, where weights are derived from key-query similarity.

## I.2  From Linear to Non-linear ICL Updates

In our linear construction (e.g., Appendix F, G), the ICL-induced update is explicitly written as:

$$\Delta x_j^{\text{ICL}} = \mathbf{PVK}^\top \mathbf{q}_j, \quad \text{and} \quad \Delta y_j^{\text{ICL}} = \mathbf{PVK}^\top \mathbf{q}_j, \tag{60}$$

which allows direct alignment with known gradient expressions. However, in non-linear attention, the presence of $\text{Softmax}(\cdot)$ introduces dynamic, input-dependent weights, breaking the closed-form linearity.

Here, we provide a more detailed derivation and justification showing that this simplification is theoretically reasonable: even under the Softmax-based attention setting, gradient descent (GD) updates can still be effectively approximated.

While Softmax attention performs a convex combination, that is, a positive weighted average, this restriction holds only in the first layer of the network. As additional non-linear attention layers are stacked and the value matrices are adjusted, the model gradually evolves beyond this constraint to realize more flexible, non-linear, and even implicitly negative-weighted, gradient-like updates.

**Derivation: Non-linear attention with Softmax approximating gradient descent.** We consider a Transformer equipped with Softmax attention and residual connections:

$$Z_{\ell+1} = Z_\ell + V_\ell Z_\ell \cdot \text{Softmax}(B_\ell X_\ell \cdot (C_\ell X_\ell)^\top), \tag{61}$$

where $Z_\ell \in \mathbb{R}^{(d+1)\times(n+1)}$ is the hidden state at layer $\ell$, $X_\ell$ denotes the covariate part (the first $d$ rows of $Z_\ell$), $B_\ell, C_\ell \in \mathbb{R}^{d\times d}$ are query/key projection matrices, and $V_\ell \in \mathbb{R}^{(d+1)\times(d+1)}$ is the value projection matrix.

To simplify the derivation, we assume

$$B_\ell = C_\ell = \frac{1}{\sigma} I_d, V_\ell = \begin{bmatrix} 0 & 0 \\ 0 & -r_\ell \end{bmatrix}, \tag{62}$$

indicating that only the label dimension is updated. Under this setup, the attention weights become

$$\text{Softmax}\left(\frac{1}{\sigma^2} X X^\top\right)_{i,j} \propto \exp\left(\frac{1}{\sigma^2} x^{(i)\top} x^{(j)}\right), \tag{63}$$

which reflects the exponentiated similarity between the query $x^{(j)}$ and context $x^{(i)}$, followed by normalization.

This forward process can be connected to functional gradient descent (FGD):

$$f_{\ell+1}(x) = f_\ell(x) + r_\ell \sum_{i=1}^{n} \left(y^{(i)} - f_\ell(x^{(i)})\right) K(x^{(i)}, x), \tag{64}$$

where $K(x, x') = \exp(x^\top x'/\sigma^2)$ is an exponential kernel and $f_\ell$ denotes the current function estimate.

We initialize the input as

$$Z_0 = \begin{bmatrix} x^{(1)} & \cdots & x^{(n)} & x^{(n+1)} \\ y^{(1)} & \cdots & y^{(n)} & 0 \end{bmatrix}, \tag{65}$$

where the first $n$ columns are training examples and the $(n+1)$-th is the query with label zero.

In the first layer, the attention module computes weights

$$\alpha_i^{(n+1)} = \frac{\exp\left(\frac{1}{\sigma^2} x^{(i)\top} x^{(n+1)}\right)}{\sum_j \exp\left(\frac{1}{\sigma^2} x^{(j)\top} x^{(n+1)}\right)} = \frac{K(x^{(i)}, x^{(n+1)})}{\sum_j K(x^{(j)}, x^{(n+1)})}, \tag{66}$$

and aggregates labels through the value matrix:

$$f_1(x^{(n+1)}) = -r_0 \sum_{i=1}^{n} \alpha_i^{(n+1)} y^{(i)}. \tag{67}$$

This step indeed performs a positive weighted average, but it serves only as a rough initial estimate of the FGD target.

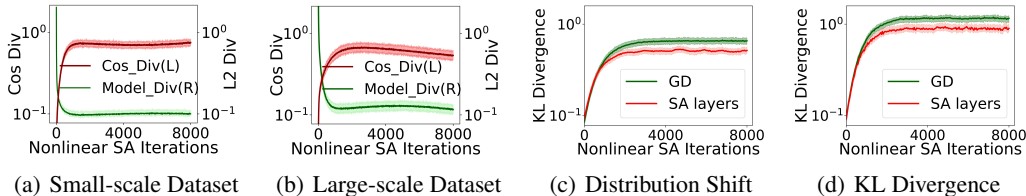

(a) Small-scale Dataset     (b) Large-scale Dataset     (c) Distribution Shift     (d) KL Divergence

Figure 6: Comparison and validation of ICL-guided strategic manipulation. (a) and (b) compare ICL and gradient-descent methods across data scales; (c) and (d) evaluate implicit gradient alignment via distribution metrics.

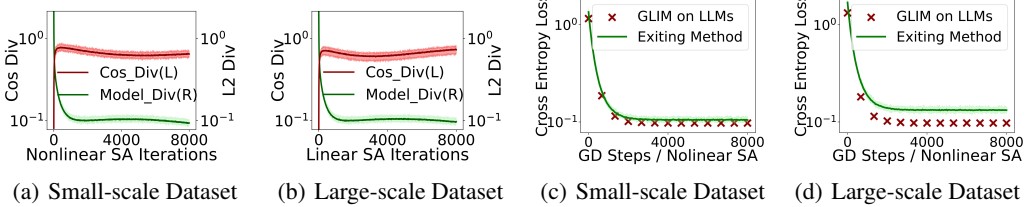

(a) Small-scale Dataset     (b) Large-scale Dataset     (c) Small-scale Dataset     (d) Large-scale Dataset

Figure 7: (a) and (b) compare ICL-guided decision rule optimization and gradient-descent methods across data scales; (c) and (d): comparison of cross-entropy losses between ICL and gradient-based methods.

**Residual updates in deeper layers.** As additional layers are stacked, residual connections accumulate prior updates and inject the residual difference $(y^{(i)} - f_\ell(x^{(i)}))$ into each non-linear attention operation. Through residual accumulation, error correction, and non-linear mixing, the update becomes

$$f_{\ell+1}(x) = f_\ell(x) + r_\ell \cdot \tau(x) \sum_{i=1}^{n} \big(y^{(i)} - f_\ell(x^{(i)})\big) K(x^{(i)}, x), \tag{68}$$

where $\tau(x)$ denotes a normalization factor induced by the Softmax scaling.

Although Softmax attention enforces positive and normalized weights locally, this constraint does not limit the model's overall expressive power. Through multi-layer stacking, tunable value projections, and residual propagation, the Transformer can effectively emulate complex update dynamics, including gradient-like steps with varying magnitude and sign. Consequently, the multi-layer non-linear attention structure can approximate a broad family of optimization trajectories, supporting the theoretical soundness of our Softmax simplification.

**Empirical and Theoretical Support.** Recent work, such as [12], demonstrates that attention layers can implement first-order optimization steps in function space, reinforcing our perspective that Transformer-based ICL can realize gradient behaviors even in the non-linear regime.

This extension is further supported by our empirical experimental validations, as shown in Figures 6 and 7.

## J   Setup Details

### J.1   Dataset Details

To evaluate our method across different domains and scales, we use a mixture of real-world and synthetic datasets, especially in **internet sector** and **financial services**, summarized as follows:

**Large-scale datasets:** *CISFraud (IEEE-CIS Fraud Detection)* [63], a large-scale transactional dataset provided by IEEE and a major international bank, containing over 1 million online payment records with identity, device, and transaction features for fraud classification. *PhiUSIIL* [55], a phishing URL detection dataset containing 134,850 legitimate and 100,945 malicious URLs, reflecting adversarial evasion scenarios in cybersecurity. *Diabetes* [66], a large-scale medical dataset with 253,680 instances, featuring demographic and clinical attributes used for type 2 diabetes risk prediction. *Synthetic* [46],

Table 3: Performance comparison across various models and datasets under Strategic and Non-Strategic settings on **large-scale datasets**

| Methods | | PhiUSIIL | CISFraud | Synthetic | Diabetes |
|---|---|---|---|---|---|
| **Existing methods (as shown in Table 1)** | | | | | |
| *Linear Model* | Strategic | $63.20_{\pm1.02}$ | $63.61_{\pm1.20}$ | $65.50_{\pm2.18}$ | $67.23_{\pm1.84}$ |
| | Non-Strategic | $57.39_{\pm0.62}$ | $56.63_{\pm1.08}$ | $60.87_{\pm2.11}$ | $61.94_{\pm1.78}$ |
| *MLP* | Strategic | $65.65_{\pm1.14}$ | $65.04_{\pm1.27}$ | $70.90_{\pm2.49}$ | $69.85_{\pm2.03}$ |
| | Non-Strategic | $59.25_{\pm0.57}$ | $59.03_{\pm1.06}$ | $65.39_{\pm2.03}$ | $63.88_{\pm2.21}$ |
| *GNN* | Strategic | $68.37_{\pm1.21}$ | $68.84_{\pm1.27}$ | $70.10_{\pm2.35}$ | $70.44_{\pm2.31}$ |
| | Non-Strategic | $59.31_{\pm0.64}$ | $60.45_{\pm1.02}$ | $65.41_{\pm2.15}$ | $64.75_{\pm2.12}$ |
| **GLIM (ours)** | | | | | |
| *DeepSeek-V3* | Strategic | $85.10_{\pm0.98}$ | $84.25_{\pm1.09}$ | $85.15_{\pm2.18}$ | $88.74_{\pm2.16}$ |
| | Non-Strategic | $78.90_{\pm1.01}$ | $78.74_{\pm1.14}$ | $80.68_{\pm2.12}$ | $81.15_{\pm1.85}$ |
| *Gemini-2.5* | Strategic | $84.17_{\pm1.03}$ | $84.41_{\pm1.09}$ | $87.18_{\pm2.20}$ | $87.60_{\pm2.54}$ |
| | Non-Strategic | $76.39_{\pm1.04}$ | $76.80_{\pm1.09}$ | $78.87_{\pm2.17}$ | $80.38_{\pm2.31}$ |
| *GPT-4o* | Strategic | $\mathbf{86.50}_{\pm0.91}$ | $\mathbf{85.51}_{\pm1.08}$ | $\mathbf{86.83}_{\pm2.35}$ | $\mathbf{89.27}_{\pm2.68}$ |
| | Non-Strategic | $\mathbf{79.14}_{\pm0.94}$ | $\mathbf{80.25}_{\pm1.10}$ | $\mathbf{81.19}_{\pm2.19}$ | $\mathbf{82.40}_{\pm2.19}$ |
| *Claude-3.7* | Strategic | $85.07_{\pm0.95}$ | $84.98_{\pm1.08}$ | $84.50_{\pm2.11}$ | $88.02_{\pm2.07}$ |
| | Non-Strategic | $78.40_{\pm0.83}$ | $77.91_{\pm1.17}$ | $78.89_{\pm2.00}$ | $80.65_{\pm2.48}$ |
| *LLama-3.3* | Strategic | $83.86_{\pm1.01}$ | $83.16_{\pm1.11}$ | $84.67_{\pm2.15}$ | $87.74_{\pm2.35}$ |
| | Non-Strategic | $76.86_{\pm0.97}$ | $75.04_{\pm1.14}$ | $76.73_{\pm2.16}$ | $79.92_{\pm2.13}$ |
| *Qwen3* | Strategic | $82.35_{\pm1.03}$ | $84.16_{\pm1.10}$ | $80.32_{\pm2.20}$ | $86.63_{\pm2.23}$ |
| | Non-Strategic | $77.29_{\pm1.08}$ | $76.26_{\pm1.16}$ | $77.34_{\pm2.20}$ | $79.10_{\pm2.00}$ |
| *Mixtral* | Strategic | $84.20_{\pm0.91}$ | $84.90_{\pm1.00}$ | $85.11_{\pm2.14}$ | $88.26_{\pm2.18}$ |
| | Non-Strategic | $77.42_{\pm0.96}$ | $77.72_{\pm1.05}$ | $78.66_{\pm2.09}$ | $80.10_{\pm2.21}$ |

a synthetic dataset generated using the PaySim simulator, which mimics mobile financial transactions and fraud patterns based on real-world data.

**Small-scale datasets:** *Adult* [4], a census dataset for predicting whether an individual's income, often used in classification tasks. *Spam* [40], a text-based dataset for binary classification of email messages as spam or not, useful for evaluating manipulation. *Credit* [79], a credit scoring dataset, used for predicting the risk of credit default in consumer finance scenarios. *German* [40], a small-scale Dataset to assess credit risk in loans from the UCI ML Repository for classification tasks. *Student* [15], a dataset includes student performance data in mathematics and Portuguese language courses.

The real-world scenes corresponding to these datasets are classified as follows:

- **Internet sector datasets**: *PhiUSIIL*, *Synthetic*, and *Spam*;
- **Financial Services datasets:** *CISFraud*, *Adult*, *Credit*, and *German*;
- **Other domain datasets:** *Diabetes* and *Student*.

## J.2    Model Selection and Configuration

Our experiments employ a variety of state-of-the-art large language models (LLMs), accessed via their respective APIs, to implement the proposed **GLIM** framework. The selected models include:

- **GPT-4o** [53]: Accessed via OpenAI's official API. This model offers enhanced reasoning and multimodal capabilities, providing robust performance in complex SC tasks.
- **DeepSeek-V3** [45]: A Chinese-English bilingual open-source LLM optimized for downstream reasoning, retrieval, and generation tasks, evaluated through DeepSeek's API platform.

Table 4: Performance comparison across various models and datasets under Strategic and Non-Strategic settings on **small-scale datasets**

| Methods | | *Credit* | *Adult* | *Spam* | *Student* | *German* |
|---|---|---|---|---|---|---|
| **Existing methods (as shown in Table 1)** | | | | | | |
| *Linear Model* | Strategic | $75.52_{\pm0.60}$ | $77.10_{\pm1.58}$ | $89.67_{\pm0.72}$ | $85.17_{\pm2.45}$ | $88.31_{\pm1.96}$ |
| | Non-Strategic | $70.73_{\pm0.31}$ | $72.16_{\pm1.62}$ | $87.52_{\pm0.58}$ | $81.82_{\pm2.34}$ | $82.07_{\pm2.01}$ |
| *MLP* | Strategic | $77.06_{\pm0.41}$ | $78.74_{\pm1.83}$ | $91.05_{\pm0.54}$ | $86.96_{\pm2.41}$ | $87.38_{\pm2.07}$ |
| | Non-Strategic | $71.50_{\pm0.33}$ | $73.57_{\pm1.55}$ | $89.01_{\pm0.69}$ | $82.05_{\pm2.16}$ | $84.82_{\pm2.45}$ |
| *GNN* | Strategic | $80.12_{\pm0.52}$ | $78.54_{\pm1.81}$ | $91.44_{\pm0.64}$ | $87.01_{\pm2.63}$ | $88.93_{\pm2.12}$ |
| | Non-Strategic | $72.27_{\pm0.34}$ | $74.37_{\pm1.57}$ | $88.13_{\pm0.71}$ | $82.90_{\pm2.58}$ | $85.14_{\pm2.37}$ |
| **GLIM (ours)** | | | | | | |
| *DeepSeek-V3* | Strategic | $89.33_{\pm0.35}$ | $86.22_{\pm1.34}$ | $94.85_{\pm0.67}$ | $89.52_{\pm2.43}$ | $91.20_{\pm2.38}$ |
| | Non-Strategic | $\mathbf{81.45}_{\pm0.41}$ | $78.77_{\pm1.33}$ | $89.31_{\pm0.68}$ | $83.32_{\pm2.57}$ | $84.97_{\pm2.06}$ |
| *Gemini-2.5* | Strategic | $84.81_{\pm0.34}$ | $85.84_{\pm1.38}$ | $94.75_{\pm0.69}$ | $88.33_{\pm2.89}$ | $90.18_{\pm2.25}$ |
| | Non-Strategic | $80.62_{\pm0.35}$ | $79.37_{\pm1.43}$ | $89.74_{\pm0.65}$ | $82.44_{\pm2.77}$ | $83.11_{\pm2.13}$ |
| *GPT-4o* | Strategic | $\mathbf{89.64}_{\pm0.27}$ | $\mathbf{91.35}_{\pm1.29}$ | $\mathbf{95.97}_{\pm0.61}$ | $\mathbf{91.61}_{\pm2.91}$ | $\mathbf{92.34}_{\pm2.45}$ |
| | Non-Strategic | $80.96_{\pm0.44}$ | $\mathbf{80.23}_{\pm1.31}$ | $\mathbf{91.28}_{\pm0.65}$ | $\mathbf{84.33}_{\pm2.79}$ | $\mathbf{85.69}_{\pm2.61}$ |
| *Claude-3.7* | Strategic | $86.51_{\pm0.31}$ | $88.58_{\pm1.51}$ | $94.50_{\pm0.66}$ | $85.92_{\pm2.54}$ | $91.07_{\pm2.33}$ |
| | Non-Strategic | $80.39_{\pm0.37}$ | $83.85_{\pm1.50}$ | $89.50_{\pm0.61}$ | $83.92_{\pm2.41}$ | $84.55_{\pm2.64}$ |
| *LLama-3.3* | Strategic | $87.58_{\pm0.30}$ | $88.70_{\pm1.41}$ | $94.30_{\pm0.64}$ | $89.44_{\pm2.93}$ | $90.68_{\pm2.19}$ |
| | Non-Strategic | $78.96_{\pm0.40}$ | $79.19_{\pm1.35}$ | $87.49_{\pm0.64}$ | $84.17_{\pm2.66}$ | $83.22_{\pm2.41}$ |
| *Qwen3* | Strategic | $87.90_{\pm0.35}$ | $88.40_{\pm1.30}$ | $95.22_{\pm0.71}$ | $88.58_{\pm2.71}$ | $90.27_{\pm2.35}$ |
| | Non-Strategic | $80.62_{\pm0.38}$ | $79.90_{\pm1.30}$ | $89.51_{\pm0.69}$ | $83.28_{\pm2.88}$ | $83.97_{\pm2.59}$ |
| *Mixtral* | Strategic | $88.24_{\pm0.29}$ | $89.04_{\pm1.33}$ | $94.12_{\pm0.63}$ | $88.92_{\pm2.63}$ | $90.84_{\pm2.42}$ |
| | Non-Strategic | $80.12_{\pm0.38}$ | $80.75_{\pm1.38}$ | $90.34_{\pm0.66}$ | $83.42_{\pm2.67}$ | $84.21_{\pm2.51}$ |

- **Claude-3.7** [3]: Provided by Anthropic via their API, Claude-3.7 emphasizes safety and alignment, making it a strong baseline for stable classification under strategic contexts.

- **Gemini-2.5** [65]: Offered by Google Cloud, Gemini models are equipped for multimodal understanding. We utilize Gemini-2.5 through Vertex AI API for tabular strategic classification tasks.

- **LLama-3.3** [49]: An open-source model by Meta, available via API endpoints and HuggingFace, used here in its instruction-tuned form (70B variant when available).

- **Mixtral** [38]: A sparse mixture-of-experts model combining multiple expert networks, suitable for dynamic contexts and scalable SC evaluations.

- **Qwen3** [13]: Provided by Alibaba Cloud, Qwen3 supports multilingual instruction following and robust handling of tabular and structured prompts. Integrated through the DashScope API.

All models are run with consistent hyperparameters across experiments. Prompt formatting is standardized to minimize variance due to stylistic differences in input-output formatting.

## J.3  Prompt Design

Effective prompt design is essential to enable in-context learning (ICL) for strategic classification (SC). We construct prompts that integrate both manipulation-aware and manipulation-agnostic settings while maintaining a consistent structure. A representative prompt example is shown below, illustrating (i) task setup, (ii) in-context examples, and (iii) batch evaluation.

**(i) Task Definition.** The prompt begins with an instruction header describing the SC setup, consistent with the theoretical formulation in Section 2.1.

> You are a strategic classification assistant. In this scenario:
>
> There are two players: a decision maker and decision subjects. - The decision maker publishes its policy (classification rule f). - The decision subjects, after observing the policy and associated costs, determine whether to strategically modify their features.
>
> Specifically, the decision maker defines a classifier mapping feature vectors to binary outcomes $y \in \{0, 1\}$. Once the rule is known, individuals may modify their features to obtain a favorable decision. Such modification incurs a cost, quantified by a cost function.
>
> The strategic manipulation rule for decision subjects is: ... (see Definition 2.1). The optimization rule for the decision maker is: ... (see Definition 2.2).
>
> Please restate your understanding of the strategic classification setting in concise terms.

**(ii) In-context Examples.** Following the instruction, a series of labeled demonstration examples (typically 12–24) are provided to simulate the adaptation process. Each example includes both the initial features and the resulting classification outcome. In the strategic condition, the feature set reflects manipulation based on our theoretical updates, while in non-strategic cases, the original features are used.

> Example 1:
>
> Initial features:
>
> - age: 34
>
> - workclass: Private
>
> - fnlwgt: 203034
>
> - education: Bachelors
>
> - education-num: 13
>
> - marital-status: Separated
>
> - occupation: Sales
>
> - relationship: Not-in-family
>
> - race: White
>
> - sex: Male
>
> - capital-gain: 0
>
> - capital-loss: 2824
>
> - hours-per-week: 50
>
> - native-country: United-States
>
> Initial result: income >50K
>
> ...
>
> Example k: ...

**(iii) Batch Evaluation.** For large-scale evaluation, prompts are programmatically generated to include a batch of test instances. Each prompt instructs the LLM to (1) apply the manipulation rule if beneficial, (2) update its decision rule accordingly, and (3) output the resulting accuracy.

> Next, I will provide you with a series of applicant cases. For each, please: 1. Apply the rules above to strategically manipulate the applicant's features if beneficial. 2. Update your decision-making rules as per the definitions above. 3. For each applicant, even after strategic manipulation, the true label should remain unchanged. Finally, report the accuracy rate under your classification rules.
>
> Test examples: ...

**Implementation Note.** In our implementation, both inner (strategic manipulation) and outer (decision adaptation) processes are unified within a single prompt, allowing the LLM to jointly simulate the two-level optimization cycle in one inference pass.

### J.4 Illustrative Example of Bi-level Optimization via GLIM

To illustrate how our GLIM framework operates within an LLM, we consider a credit approval task as an example.

Given several in-context examples of applicants with varying financial histories and approval outcomes, the LLM implicitly infers a decision rule. Some features may exert a stronger influence (e.g., recent payment behavior), while others contribute less, shaping the internal classification boundary through attention dynamics.

When a new agent is presented, the LLM not only predicts the approval outcome but also implicitly anticipates how the applicant might strategically manipulate certain features (e.g., reducing overdue counts or increasing recent payments) to receive a favorable decision.

Through repeated exposure to strategically manipulated examples in the prompt, the LLM implicitly adjusts its decision rule toward a more stable and robust form. This corresponds to the outer-stage optimization, while the simulated feature changes capture the inner-stage manipulation, together completing the bi-level strategic classification process.

This example simulates both stages of strategic classification through forward-only inference, all without parameter tuning, relying entirely on in-context learning.

### J.5 Implementation Details

The experimental pipeline is implemented in Python, with integration across multiple LLM APIs. Our infrastructure supports large-scale evaluation while ensuring reproducibility. Official SDKs (e.g., openai, anthropic, google-cloud-aiplatform) are used to interface with model APIs. All keys are securely stored via encrypted environment variables. A modular prompt generator selects appropriate examples and formats based on dataset type, manipulation setting, and model constraints. This allows easy extensibility to new datasets or model variants.

## K Additional Experimental Results

We apply our proposed ICL-based approach, GLIM, across multiple large language model APIs and document the detailed results in Tables 3 and 4.

In this comparison, we note that the baseline models used in our experiments, such as linear models and shallow neural networks, are optimized through traditional training procedures involving parameter updates. In contrast, GLIM operates in a zero-update regime, leveraging the inherent in-context reasoning capabilities of pre-trained LLMs. This distinction reflects a difference in mechanism rather than in model capacity.

# L  Discussion on Our Method

## L.1  Prompting Cost for Strategic Manipulation of Agent

One potential concern when adapting strategic classification (SC) to large language models (LLMs) lies in the cost of interaction, particularly the computational and structural overhead associated with in-context learning (ICL). In classical SC literature, agent-side manipulation cost is often modeled as a transformation cost over input features, such as Mahalanobis or $L_p$ norms. In contrast, the LLM setting introduces a new dimension: the cost of prompts for agents.

In our framework, however, we intentionally abstract away the prompting cost by adopting a perceptual prediction view of agent behavior, consistent with recent theoretical perspectives in strategic learning [50, 61]. From this perspective, agents are modeled as cognitive entities who respond by adjusting their feature representations. The prompt cost, being a fixed system, level expense unrelated to individual feature manipulation, does not influence the agent's strategic calculus.

Moreover, in real-world deployments, prompt construction and transmission are typically handled by system infrastructure or shared communication protocols, incurring negligible marginal cost for individual agents. In contrast, modifying one's features (e.g., improving test scores, altering financial statements) entails significant personal effort or risk. Therefore, although we acknowledge that prompting costs may be relevant in certain system-level analyses, they play no substantive role in the agent-level incentive structure we seek to model. Hence, we choose to omit prompt cost from our formal analysis.

## L.2  Discussion on Theory-Practice Divergence at Scale

As shown in Figures 3(b) and 6(b), the cosine similarity tends to decrease over iterations, indicating a divergence between our theoretical analysis and the empirical results at scale. This divergence may be attributable to factors such as model capacity, data distribution shifts, or nonlinearities in real-world tasks.

We acknowledge the existence of this divergence and further analyze this phenomenon in future work to better understand the conditions under which such divergences occur.

## L.3  Discussion on API Cost Analysis of GLIM

A key practical limitation of our approach is its reliance on proprietary large language models that are accessed via commercial APIs, such as OpenAI GPT-4o and DeepSeek. Unlike traditional machine learning models, which can be trained or deployed locally with a fixed hardware budget, our method depends on repeated calls to remote LLM API services.

In our experiments, we observed that the total inference cost is not constant but grows with the number of API requests. Each additional example increases the total token count in a roughly proportional manner, resulting in higher overall expenses. This observation highlights the need to account for API-related cost structures when designing systems that involve frequent or large-scale model queries. Such variability is inherent to current commercial LLM platforms and represents a fundamental constraint for real-world deployment.

To mitigate this limitation, several directions can be explored. One potential strategy is to design more cost-efficient prompt engineering methods that reduce the number of required API calls without compromising performance. However, in practice, the dynamic nature of agent distributions and context diversity makes strict control over prompt volume challenging. Another promising direction is the development of hybrid systems, where LLM-based reasoning is selectively applied to complex or ambiguous cases, while lightweight local models handle routine or latency-sensitive requests. This hybrid paradigm could significantly reduce dependence on commercial APIs while maintaining flexibility and adaptability.

Addressing these practical constraints will be an important avenue for future work, as we aim to optimize the trade-off between the flexibility of LLM-powered reasoning and the cost-effectiveness required for sustainable large-scale deployment.

