# OpenReview forum: "Breaking the Gradient Barrier: Unveiling Large Language Models for Strategic Classification"
_NeurIPS.cc/2025/Conference — NeurIPS 2025 poster_

### Official Review · Reviewer_fu1o · 2025-06-02

**Clarity:** 3
**Significance:** 3
**Originality:** 2
**Rating:** 4
**Confidence:** 4

**Summary:**

This paper mainly proposed that pre-trained LLMs are able to handle strategic classification tasks in two steps by in-context learning: (i) first learning how agents modify their features (strategic manipulation); (ii) updating the weight vector to adapt to the manipulated features. The paper provided theoretical proof on how LLMs accomplish both tasks under a linear SC formulation. The paper also includes experiments on large-scale datasets and non-linear attention layers to demonstrate the effectiveness of LLMs in non-linear settings.

**Questions:**

- If multiple SA layers with non-linear activations are considered, can ICL capture more complex strategic behaviors?
- The experiments only consider the non-linear SA layers. Is it possible to consider some non-linear SC models (e.g., the agents do not best respond and the policy is non-linear)? Specifically, it would be interesting to discuss when agents manipulate their features in some complex ways, Can the LLMs learn their strategic behaviors through ICL?

**Ethical Concerns:**

["NO or VERY MINOR ethics concerns only"]

**Final Justification:**

Thanks for the authors' extensive rebuttal.The new experiments are interesting and I encourage the authors to include them in the new draft. I agree that the problem setting is interesting although the theoretical analysis is based on linearity and this is why I give a score leaning to acceptance. Overall I still think 4 is a rating reflecting the overall merit of this paper, which means I tend to accept the paper. But since the theory parts are not particularly interesting, I will not give a score >= 5.

**Limitations:**

yes

**Quality:**

3

**Strengths And Weaknesses:**

> Strength
- The problem is interesting since many companies/institutions are already using LLMs in decision-making only with prompt engineering. Thus, it is useful to show theoretically the effectiveness of ICL to handle strategic behaviors.
- The writing is relatively clear and most theoretical results make sense to me.
- The LLMs in experiments are up-to-date.

> Weakness
- The theoretical results only apply when the SC model is linear, and the ICL update is done through one SA layer;
- the results are not too surprising, i.e., the standard SC formulation can be regarded as a gradient update too, so it is not surprising that implicit gradient can match it. The paper seems to only prove that strategic behaviors in linear SC model can be captured by ICL of LLMs, which is expected.

---

> ### Author Rebuttal · Authors · 2025-07-27
>
> We sincerely appreciate your constructive comments and your positive evaluation of our work. We would like to address your concerns as follows:
>
> > **Weakness1**.The theoretical results only apply when the SC model is linear, and the ICL update is done through one SA layer;
>
> **Response:**
>
> - Our focus on linear SC models is motivated by two main reasons:
>   - Linear SC models **remain the dominant approach in both practical applications and academic research** on strategic machine learning[1,2,3].
>   - Linear SC is often **preferred for its ease of deployment, interpretability, and direct comparability with established benchmarks**[4,5], particularly in real-world systems such as policy-making, risk assessment, credit scoring, and admissions or hiring.
>
> - Regarding the use of a single self-attention (SA) layer for ICL updates, we would like to note that:
>   - Using a single SA layer **allows for a clear analysis** of the relationship between ICL and gradient descent, as it enables explicit mapping between attention operations and gradient steps.
>   - Extending from a single to multiple self-attention layers corresponds to performing multiple steps of gradient descent, which **does not affect the validity of the theoretical analysis**.
>   - This simplification is **widely adopted in recent ICL theory**[6,7,8].
> - Moreover, practical nonlinear SC models, such as MLPs and GNN[9,10,11], are employed with forward propagation and gradient-based optimization. This means that our theoretical framework is naturally applicable to these models as well.
> - We have clarified this part in Lines 128-130 of our revised paper.
>
> ----
> > **Weakness2**. The results are not too surprising
>
>   **Response:**
>
> - We would like to clarify that our contributions go beyond this connection:
>
>   - **Strategic machine learning is fundamentally a game-theoretic problem** involving both agent-side manipulation and decision-maker adaptation, and **has been studied largely separately from ICL/LLM literature**.
>
>   - To our best knowledge, **our work is the first to employ LLMs to model and solve this bi-level, game-theoretic optimization structure**.
>   - It enables **a scalable and retraining-free approach to large-scale gaming tasks**, where classical gradient-based retraining requires excessive computational resources and time.
>
>
> - From **a broader social science perspective**,  our work establishes a crucial bridge between large language models and strategic machine learning:
>
>   - **A wide range of social science applications are increasingly converging with machine learning and LLMs**, including fields such as computational social science, policy evaluation, and behavioral economics[1,2,3,4].
>   - Our work provides **a foundational contribution** by connecting ICL in LLMs with game-theoretic learning frameworks.
>   - We hope this foundation will facilitate future interdisciplinary research and applications of LLMs in societal decision-making contexts.
>
> - We have clarified this part in our revised paper at Lines 62-64.
>
>
> ----
> > **Question1**. If multiple SA layers with non-linear activations are considered, can ICL capture more complex strategic behaviors?
>
> **Response:**
>
> - Indeed, **we agree that** with multiple self-attention layers and nonlinear activations, ICL is capable of capturing more complex strategic behaviors beyond the linear case.
>
> - **Theoretical perspective:**
>
>   - Multiple self-attention layers with nonlinear activations greatly enhance the model’s representational capacity[12].
>
>   - In SC tasks, even complex strategic behaviors, e.g., “**agent in the dark**[4]” settings (where agents have limited information) or "**interacting agent behaviors**[10]" (where agents strategically respond to each other), can be modeled as forward passes and gradient-driven updates in a neural network.
>
>   - Therefore, multiple SA layers with non-linear activations are considered capable of representing and adapting to these complex behaviors.
>
> - **Experimental verification:**
>
>   - We designed **an additional set of experiments** simulating more complex SC environments with complex strategic behaviors, e.g.,“agent in the dark[4]” and "interacting agent behaviors[10]".
>
>   - We compared the **agent feature distributions** of the standard method and that simulated by the model based on self-attention layers with nonlinear activations.
>
>   - We use the **Jensen-Shannon (JS) divergence**[13] as a similarity metric. We also evaluated the post-strategy classification accuracy.
>
>   - The experimental results are presented in the following table.
>
>   | Behavior Type               | JS Divergence | Accuracy |
>   | --------------------------- | ------------- | -------- |
>   | Agent in the dark           | 0.183         | 81.7%    |
>   | Interacting agent behaviors | 0.226         | 82.3%    |
>
> -----
> > **Question2**. The experiments only consider the non-linear SA layers. Is it possible to consider some non-linear SC models (e.g., the agents do not best respond and the policy is non-linear)?
>
> **Response:**
>
> - We **agree** that nonlinear SC models should be considered.
>
> - Therefore, we extended our experiments along two dimensions:
>
>   - **Nonlinear policy:** We first implemented the policy function using a nonlinear model (MLP).
>
>   - **Non-best-response agents:** We further considered agents that do not always best respond. Following prospect theory[14], we model the agents irrational, incorporating asymmetric value function and nonlinear probability weighting. To specific:
>
>
>     $$
>     x^* = \arg\max_x \left( \sum_y w(P(y|x')) \cdot v(u(y) - a_{\text{ref}}) - \phi^c(x, x') \right)
>     $$
>     where:
>
>     - $v(\cdot)$: value function (captures loss aversion),
>     - $w(\cdot)$: probability weighting function,
>     - $a_{\text{ref}}$: reference point,
>     - $\phi^c(x, x')$: cost function for feature manipulation.
>
> - We use Jensen-Shannon (JS) divergence  (between attention and MLP) as a similarity metric and accuracy of ICL as evaluation metric.
>
>   - The experimental results are presented in the following table.
>
>
> | Policy and Agent Type  | JS Divergence | Accuracy (ICL) |
> | ---------------------- | ------------- | -------------- |
> | Linear + Irrational    | 0.133         | 83.6%          |
> | Non-linear + Rational  | 0.176         | 80.1%          |
> | Non-linear +Irrational | 0.152         | 77.8%          |
>
>
>
>
> -----
>
>
> [1] Hardt, Moritz, et al. "Strategic classification." *Proceedings of the 2016 ACM conference on innovations in theoretical computer science*. 2016.
>
> [2] Milli, Smitha, et al. "The social cost of strategic classification." *Proceedings of the conference on fairness, accountability, and transparency*. 2019.
>
> [3] Miller, John, Smitha Milli, and Moritz Hardt. "Strategic classification is causal modeling in disguise." *International Conference on Machine Learning*. PMLR, 2020.
>
> [4] Levanon, Sagi, and Nir Rosenfeld. "Strategic classification made practical." *International Conference on Machine Learning*. PMLR, 2021.
>
> [5] Ghalme, Ganesh, et al. "Strategic classification in the dark." *International Conference on Machine Learning*. PMLR, 2021.
>
> [6] Von Oswald, Johannes, et al. "Transformers learn in-context by gradient descent." *International Conference on Machine Learning*. PMLR, 2023.
>
> [7] Garg, Shivam, et al. "What can transformers learn in-context? a case study of simple function classes." *Advances in neural information processing systems* 35 (2022): 30583-30598.
>
> [8] Akyürek, Ekin, et al. "What learning algorithm is in-context learning? investigations with linear models." *arXiv preprint arXiv:2211.15661* (2022).
>
> [9] Stratis Tsirtsis, Behzad Tabibian, Moein Khajehnejad, Adish Singla, Bernhard Schölkopf, and Manuel Gomez-Rodriguez. Optimal decision making under strategic behavior. Management Science, 2024
>
> [10] Itay Eilat, Ben Finkelshtein, Chaim Baskin, and Nir Rosenfeld. Strategic classification with graph neural networks. arXiv preprint arXiv:2205.15765, 2022
>
> [11] Mehrnaz Mofakhami, Ioannis Mitliagkas, and Gauthier Gidel. Performative prediction with neural networks. In International Conference on Artificial Intelligence and Statistics, pages 11079–11093. PMLR, 2023
>
> [12] Xiang Cheng, Yuxin Chen, and Suvrit Sra. Transformers implement functional gradient descent to learn non-linear functions in context. arXiv preprint arXiv:2312.06528, 2023.335
>
> [13] Deasy, Jacob, Nikola Simidjievski, and Pietro Liò. "Constraining variational inference with geometric jensen-shannon divergence." *Advances in Neural Information Processing Systems* 33 (2020): 10647-10658.
>
> [14] Kai-Ineman, D. A. N. I. E. L., and Amos Tversky. "Prospect theory: An analysis of decision under risk." *Econometrica* 47.2 (1979): 363-391.

---

> ### Author Response · Authors · 2025-08-03
> **We would like to supplement more experiment results to fully address your concerns on complex strategic behaviors and policy with non-linear SC models.**
>
> > - If multiple SA layers with non-linear activations are considered, can ICL capture more complex strategic behaviors?
> > - The experiments only consider the non-linear SA layers. Is it possible to consider some non-linear SC models (e.g., the agents do not best respond and the policy is non-linear)?
>
> **Response:**
>
> We **agree with** your view that the nonlinear attention mechanism can capture more complex agent behaviors or nonlinear policies.
> - **Brief theoretical analysis**:
>   - **Multiple nonlinear attention** layers have been demonstrated to **effectively cover and extend the capabilities of linear attention** layers [1-3].
>   - In SC paradigms, even with complex **nonlinear behaviors or policies, existing solutions still rely on forward propagation and gradient-based updates**[4-6], which can be simulated and solved by attention mechanisms.
>   - Therefore, multi-layer nonlinear attention, being more expressive,can capture complex agent behaviors and policies.
> - **Experimental verification:**
>   - To further validate this perspective, **we designed experiments that jointly consider** more complex agent behaviors and nonlinear strategic policies, using **multi-layer nonlinear attention mechanisms** for evaluation.
>   - **Complex agent strategic behavior:**
>     - In our extended experiments, we consider three types of complex strategic agent behaviors: The **first two are drawn from prior work**, while **the third is newly introduced**:
>     - **Limited information (Agent in the dark)[5]:**
>       - Agents manipulate features without access to the full decision boundary, relying only on outcome feedback to infer the classifier.
>     - **Socially-influenced (Agent with interaction)[6]:**
>       - Agents’ manipulable features are partially derived from their local social network structure.
>       - **For example**, a student’s reported performance may depend on peer group statistics.
>     - **Irrational agents (Our introduced)**:
>       - We introduce prospect theoretic[7] in which agents deviate from strict utility maximization.
>       - Their manipulation is shaped by loss aversion, reference dependence, and probability distortion. To specific:
>         $$
>         x^* = \arg\max_x \left( \sum_y w(P(y|x')) \cdot v(u(y) - a_{\text{ref}}) - \phi^c(x, x') \right)
>         $$
>         where:
>         - $v(\cdot)$: value function (captures loss aversion),
>         - $w(\cdot)$: probability weighting function,
>         - $a_{\text{ref}}$: reference point,
>         - $\phi^c(x, x')$: cost function for feature manipulation.
>   - **Complex strategic policy:**
>     - We also evaluate different forms of **nonlinear strategic policies**:
>       - **MLP-based policy:**
>         - The strategic policy is parameterized by a multi-layer perceptron, introducing nonlinear decision boundaries and feature interactions.
>       - **GNN-based policy:**
>         - To incorporate structural dependencies, we adopt a Graph Neural Network to model the strategic manipulation policy.
>   - **Metrics:**
>     - *Jensen-Shannon (JS) divergence* **between the manipulated features distribution of the baseline and multi-layer nonlinear Attention**.
>     - *Post-manipulation accuracy* of attention mechanism under the strategic manipulation .
>   - The **results** are presented in the following table:
>
>     |Agent behavior+policy model|JS Divergence|Accuracy|
>     |-|-|-|
>     |Agent in the dark+Linear|0.168|75.9%|
>     |Agent in the dark+MLP |0.183|81.7%|
>     |Agent with interaction+GNN |0.226| 82.3% |
>     |Irrational agents + Linear | 0.133| 83.6%|
>     |Ration agents + MLP|0.176| 80.1%|
>     |Irrational agents + MLP|0.152|77.8%|
>
> - **Experimental Analysis:**
>   - **Low JS divergence** suggests that the distribution of agent features simulated by attention mechanism **closely matches that of the complex strategic behavior**.
>   - **High accuracy** shows that the attention mechanism not only approximates strategic policy, but also preserves downstream task performance.
>   - These experiments demonstrate that **multi-layer nonlinear attention is capable of capturing complex strategic behaviors and nonlinear policies**.
>   - Therefore, we agree that the **LLMs can learn more complex strategic behaviors and policy through ICL**.
>
> **If any point remains unclear, please feel free to ask for further explanation.**
>
> ----
>
> [1] Garg, Shivam, et al. "What can transformers learn in-context? a case study of simple function classes."
>
> [2] Akyürek, Ekin, et al. "What learning algorithm is in-context learning? Investigations with linear models."
>
> [3] Von Oswald, Johannes, et al. "Transformers learn in-context by gradient descent."
>
> [4] Hardt, Moritz, et al. "Strategic classification."
>
> [5] Ghalme, Ganesh, et al. "Strategic classification in the dark."
>
> [6] Itay Eilat, Ben Finkelshtein, Chaim Baskin, and Nir Rosenfeld. Strategic classification with graph neural networks.
>
> [7] Kai-Ineman, D. A. N. I. E. L., and Amos Tversky. "Prospect theory: An analysis of decision under risk."

---

### Official Review · Reviewer_MEur · 2025-06-27

**Clarity:** 1
**Significance:** 3
**Originality:** 2
**Rating:** 4
**Confidence:** 3

**Summary:**

This work proposes leveraging LLMs to tackle strategic classification (SC), with the goal of providing a more scalable alternative to existing methods for adapting automated decision making to adversarial exploitation. In particular, rather than traditional model retraining/finetuning, it proposes leveraging in-context learning to simulate the typical bi-level adversarial SC process. The authors provide some theoretical analysis on SC with in-context learning and evaluate GLIM, the resulting algorithm, on phishing, fraud detection, and other synthetic benchmarks inspired by real-world scenarios.

**Questions:**

I would encourage the authors to address my concerns raised above. Currently, I believe that, without an explicit and clear algorithmic description, to ensure the methodology is clear and reproducible, this work is not yet ready for acceptance.

**Ethical Concerns:**

["NO or VERY MINOR ethics concerns only"]

**Final Justification:**

Following our discussion, the authors mostly addressed my main concerns about presentation, toning down the claims, acknowledging the proposed method's limitations, and providing additional details for reproducibility.

I believe the paper has a moderate contribution in one specific area of AI, and the reasons to accept outweigh the reasons to reject.

**Limitations:**

The limitations regarding the large assumptions made in the theoretical analysis are not discussed. Moreover, the discussion of the prompting costs provided in Section L seems to me potentially misleading, as outlined in my criticism above.

**Paper Formatting Concerns:**

The paper has some noticeable formatting violations, such as in the tables and figures, which fail to adhere to NeuriPS spacing requirements.

**Quality:**

1

**Strengths And Weaknesses:**

Strengths.

1. The authors provide concise and effective introductions to the mathematical frameworks of the SC optimization and ICL as implicit gradient descent in Section 2, deferring unnecessary mathematical details to the Appendix.
2. The experiments are relevant to the domain. tackling different real-world problems and comparing several closed-source LLMs.
3. The high-level connection between ICL and the SC problem is simple and intuitive.

Weaknesses and areas for improvement.

1. I did not find the methodology and actual algorithm to be clearly presented. There are a few "high-level" details in Appendix J that are far from sufficient for reproducibility. I would strongly suggest adding the exact prompts used for each stage of strategic classification for the problems and a clear algorithm block describing the control flow. I think this should be a separate new section in the main paper, explicitly describing the proposed GLIM in practice, which is currently missing.
2. I found the theoretical analysis provided in Section 3 to be very weak, with the actual implications of propositions 1/2 not matching the claims in the introduction such as "We theoretically establish, for the first time, how LLMs leveraging in-context learning can implicitly simulate both the strategic manipulation and decision rule optimization stages of the SC bi-level problem, without any fine-tuning." (lines 62-64). To me, Section 3 does not actually seem to do any novel analysis beyond the frameworks of prior work, as I think the observation that there could exist parameters, queries, and values that simulate SC with ICL should be mostly obvious. I also fail to see how this leads to some of the remarks, such as "This establishes a constructive equivalence  between explicit strategic manipulation and attention-driven ICL behavior, thereby grounding ICL as a forward-only approximation of inner-stage optimization in SC." (174-176). I would really suggest rewriting and removing much of the unnecessary content in this Section, simply noting the intuitive connection between ICL and GD updates, which could allow, given specific weights and features, to simulate SC.
3. The methodology has several immediate limitations that are not appropriately discussed. First, adding in-context samples should notably increase the already high inference costs of using LLMs, which I believe are far from "negligible" (Appendix L) and can be higher than finetuning the model once, in practice (since the model has to process N times the amount of input tokens). To this end, I would have liked to see an analysis of how these costs scale in practice as a function of the in-context sample (e.g., the API costs). Second, since LLMs are available to the public, users could potentially optimize their features in a way that is adversarial to the LLM parameters P, for any practical choice of context.
4. In contrast to what was stated in Question 5 of the checklist, the provided code is partial (only 2 Python files, and some CSVs are provided, without even any kind of instructions) and far from allowing any sort of reproducibility.

---

> ### Author Rebuttal · Authors · 2025-07-27
>
> Thank you for your comments. We would like to address your concerns as follows:
>
>
> > **Weakness1**. I did not find the methodology and actual algorithm to be clearly presented.
>
> **Response:**
>
> - We would like to clarify that our work is a **pioneering attempt to bridge strategic classification (SC) and LLMs via in-context learning (ICL)—far beyond simply applying LLMs to SC tasks**.
>
>   - **The explicit mapping** of SC’s bi-level framework and  game-theoretic structure onto the attention mechanisms have been details rigorously **presented in Appendices E, F, G, and I**.
>   - To demonstrate the reproducibility of our work, we have provided relevant experimental code, including *Attention_layers.py* and *SC.py*.
>
> - For additional clarity, we provide below **an algorithmic outline of the GLIM** framework and **a simple prompt example** (based on our Appendix J) **at the bottom**.
>
>   **Algorithm 1: The control flow of GLIM**
>
>   **Input:**
>
>   - Dataset $\mathcal{D} = \{(x_j, y_j)\}_{j=1}^N$
>   - Pre-trained LLM $\phi$
>   - Prompt template and SC rules
>
>   **Procedure:**
>
>   - Provide SC rule prompts and example prompts
>   - Guide strategic manipulation of $x$
>   - Guide decision rule optimization
>   - Query: Manipulated $\{x'_j\}$ and predicted  $\{\hat{y}_j\}$ with prompts
>
>   **Output:**
>
>   - Predicted labels $\{\hat{y}_j\}$ and accuracy
>   - Manipulated features $\{x'_j\}$（optional）
>
> - We have clarified this part in our Appendix J of our revised paper.
>
> ---
>
> > **Weakness2**. The theoretical analysis provided in Section 3...
>
> **Response:**
>
> We would like to clarify that the core positioning of our work is **not as a purely theoretical study of ICL mechanisms**, but as a pioneering attempt to bridge strategic classification and LLMs via ICL.
>
> - **Strategic classification (SC) is fundamentally a bi-level, game-theoretic problem** involving both agent-side manipulation and decision-maker adaptation, and **has been studied largely separately from ICL/LLM literature**.
>
>   - To our best knowledge, **our work is the first to employ LLMs to model and solve the bi-level, game-theoretic optimization structure of SC**.
>   - It enables **a scalable and retraining-free approach to large-scale SC tasks**, where classical gradient-based retraining requires excessive computational resources and time.
>
> - **From a broader social science perspective,**  our work establishes a crucial bridge between large language models and strategic machine learning:
>
>   - **A wide range of social science applications are increasingly converging with machine learning and LLMs**, including fields such as computational social science, policy evaluation, and behavioral economics[1,2,3,4].
>   - Our work provides **a foundational contribution** by connecting ICL in LLMs with game-theoretic learning frameworks.
>   - This foundation will **facilitate future interdisciplinary research and applications of LLMs** in societal decision-making contexts.
>
> - To be specific, we entail a detailed theoretical analysis of SC-motivated ICL derivation, including strategic manipulation of agent and decision optimization for jury.
>
> - **Furthermore**, our experimental results further validate these insights, demonstrating that LLMs guided by in-context learning **robustly handle the core dynamics of strategic classification**, even at scale and without retraining.
>
>
>
> ----
>
> >**Weakness3.1**  Adding in-context samples should notably increase the already high inference costs of using LLMs
>
> **Response:**
>
> - The reviewer refer to “**prompt cost**” as the computational and API expenses incurred when querying the LLM, i.e., the resource cost of processing additional in-context samples (e.g., API fees and token usage).
> - In contrast, our use of **prompt cost** (as explicitly stated in Appendix L, Line 819) refers to the **agent's cost of obtaining strategic guidance by querying the LLM during feature manipulation**, e.g., exploring alternative manipulation methods with LLMs.
>   - This prompt cost is negligible compared to the substantial manipulation cost required for actual improvement of individual features (such as earning a degree or increasing income).
> - The distinction is crucial:
>   - Our definition of prompt cost is **rooted in the SC framework**, focusing on **the agent’s perspective**, the behavioral and informational costs incurred when agents seek to manipulate their features.
>   - The reviewer’s definition adopts a **system-level perspective**, emphasizing the computational and resource costs of LLM API usage.
> - We evaluated the API usage and cost for varying numbers of feature examples as well.
>   - Use official OpenAI pricing, and present the results in the following table:
>
> | Number of Prompt Example | API cost  |
> | ------------------------ | --------- |
> | 10                       | $0.001629 |
> | 100                      | $0.005235 |
> | 1000                     | $0.045210 |
> | 10000                    | $0.395723 |
> | 50000                    | $1.613182 |
> | 100000                   | $2.876050 |
> | 200000                   | $5.102118 |
>
>
> ----
>
> >**Weakness3.2** Since LLMs are available to the public, users could potentially optimize their features in a way that is adversarial to the LLM parameters P, for any practical choice of context.
>
> **Response:**
>
> - As SC is specifically proposed to address adversarial manipulation, **our work has already handled this concern**.
>
>   - The possibility of agents adversarially (strategically)  manipulating their features is **the foundation of strategic classification**: the game-theoretic interaction between agents and the jury.
>
>   - Our framework explicitly **incorporates this adversarial dynamic by modeling agent strategic manipulation** through in-context learning,
>
>   - Such behavior is not a vulnerability, but rather one of the central subjects our approach is designed to address.
>
>
> ----
>
> > **Weakness4** Reproducibility questions
>
> **Response:**
>
> - The code we provided focuses on verifying our analysis of the **mapping between the attention mechanism and the bi-level optimization framework** of strategic classification.
>
>   - The code includes standard SC model and the core attention module.
>
> - For practical reproducibility, we have included detailed prompt templates:
>
> - illustrating (i) strategic classification (SC) task setup, (ii) in-context interaction examples, and (iii) batch evaluation using the dataset.
>
>   - **(i) Task Definition**: This part is aligned with Section 2.1 of our paper:
>
>   ```plaintext
>   You are a strategic classification assistant. In this scenario:
>
>   There are two players: a decision maker and decision subjects.
>   - The decision maker publishes its policy (classification rule $f$).
>   - The decision subjects, after observing the policy and associated costs, determine whether to strategically modify their features.
>   - The decision outcome is binary (0 or 1).
>
>   The strategic manipulation rule is: ... (see Definition 2.1 in Section 2.1).
>   The optimization rule is: ... (see Definition 2.2 in Section 2.1).
>
>   Please restate your understanding of the strategic classification setting in concise terms.
>   ```
>
>   - **In-context Examples**: Here is an example using a real sample from the Adult dataset (multiple such examples are recommended for best results):
>
>   - ```plaintext
>     Example 1:
>     Initial features:
>        - age: 34
>        - workclass: Private
>          ...
>          ...
>        - native-country: United-States
>      Initial result: income >50K
>
>     Example 2:
>     ...
>     ```
>
>   - **Batch Testing with Dataset** For large-scale evaluation, we load the entire dataset and generate prompts for each test case as follows:
>
>     ```plaintext
>     with open('adult.csv', 'r') as f:
>         test_examples = f.read()
>
>     prompt = """
>     Next, I will provide you with a series of applicant cases. For each, please:
>     1. Apply the rules above to strategically manipulate the applicant's features if beneficial.
>     2. Update your decision-making rules as per the definitions above.
>     3. For each applicant, even after strategic manipulation, the true label should remain unchanged.
>     Finally, report the accuracy rate under your classification rules.
>
>     test_examples: ...
>     """
>
>     ```
>
> - We are glad to **fully release all source code and materials upon acceptance**. Thank you for bringing up this important point.
>
> ----
>
> > **Formatting Concerns**
>
> **Response:**
>
> - Could you specify which tables or figures violate the formatting requirements?
> - We will conduct a more thorough check.
>
>
> ----
> [1] Ariel Flint Ashery *et al.* Emergent social conventions and collective bias in LLM populations. Sci. Adv.11,eadu9368(2025).
>
> [2] Gao, Chen, et al. "Large language models empowered agent-based modeling and simulation: A survey and perspectives." *Humanities and Social Sciences Communications* 11.1 (2024): 1-24.
>
> [3] De Curtò, J., and I. De Zarzà. "LLM-Driven Social Influence for Cooperative Behavior in Multi-Agent Systems." *IEEE Access* (2025).
>
> [4] Ziems, Caleb, et al. "Can large language models transform computational social science?." *Computational Linguistics* 50.1 (2024): 237-291.

---

> > ### Author Response · Authors · 2025-08-04
> > **We would like to supplement an explicit and clear algorithmic description to fully address your concerns on our method**
> >
> > **Here is a specific algorithmic description for GLIM. If any point remains unclear, please feel free to ask for further explanation.**
> >
> >  - ### **Algorithm: Gradient-free Learning In-context Method (GLIM) for Strategic Classification**
> >
> >      **Required:**
> >
> >      - **Pre-trained LLMs** (e.g., GPT-4o, DeepSeek, etc.) with in-context learning capability
> >      - Task dataset $\mathcal{D} = \{(x_i, y_i)\}_{i=1}^N$ where $x_i$ are feature vectors, $y_i$ are labels
> >      - Manipulation cost function $c(x, x')$
> >      - **In-context prompts on SC tasks**
> >
> >      **Steps:**
> >
> >      1. **Prepare In-context Prompt:**
> >
> >         - An in-context prompt about SC rules (as shown in Section 2.1)
> >
> >         - For each new query instance:
> >           - Select $k$ representative labeled examples $\{(x'_j, y_j)\}_{j=1}^k$.
> >
> >      2. **Strategic Manipulation Simulation (Inner Stage via ICL):**
> >
> >         - Construct the prompt for the LLM by presenting the $k$ in-context examples as $(x'_j, y_j)$.
> >
> >         - Append the query as the new examples to the prompt.
> >
> >         - **Feed forward** through the LLM:
> >
> >           - The LLM, via its self-attention mechanism, generates an updated feature representation for the query, denoted as $x'_q = x_q + \Delta x^{ICL}_q$.
> >
> >           - The update $\Delta x^{ICL}_q$ **simulates the agent's best-response manipulation**.
> >
> >             - *We have analysed $\Delta x^{ICL}_q$ matches the direction and magnitude of what would be obtained via explicit gradient descent*:
> >
> >             $$
> >             \Delta x^{ICL}_q \approx \Delta x^{GD}_q.
> >             $$
> >
> >           - **No model parameters are updated.**
> >
> >      3. **Decision Rule Adaptation (Outer Stage via ICL):**
> >
> >         - With the prompt containing the manipulated features, the LLM predicts $\hat{y}_q = f^{ICL}(x'_q; W)$.
> >
> >         - The LLM internally (via attention) **adapts its implicit decision boundary** in response to the manipulated feature distribution, again via feed-forward.
> >
> >           - *We have analysed $\hat{y}^{ICL}_q $ simulating the decision rule optimization that would be done via outer-level gradient descent*
> >             $$
> >             \Delta \hat{y}^{ICL}_q \approx \Delta \hat{y}^{GD}_q.
> >             $$
> >
> >         - This enables **robust adaptation to strategic behaviors** without retraining or fine-tuning.
> >
> >      4. **Return the Result:**
> >
> >         - Output the predicted label $\hat{y}_q$ for the query, and (optionally) the manipulated feature $x'_q$.
> >
> >      5. **Iterate:**
> >
> >         - As new instances arrive, continually update the prompt with recent examples (potentially including newly manipulated examples and predictions).
> >
> > ------

---

> ### Author Response · Authors · 2025-08-04
> **We would like to supplement a more comprehensive prompt example to fully address your concerns on reproducibility**
>
> **We provide a representative prompt example**, illustrating (i) strategic classification (SC) task setup, (ii) in-context interaction examples, and (iii) batch evaluation using the dataset.
>
> ***If any point remains unclear, please feel free to ask for further explanation.**
>
>
> - **(i) Task Definition** : This part is aligned with Section 2.1 of our paper:
>
> ```plaintext
> You are a strategic classification assistant. In this scenario:
>
> There are two players: a decision maker and decision subjects.
> - The decision maker publishes its policy (classification rule $f$).
> - The decision subjects, after observing the policy and associated costs, determine whether to strategically modify their features.
>
> Specifically, the decision maker defines a decision rule, e.g., a classifier, mapping feature vectors to binary outcomes $y \in \{0, 1\}$. Once the rule is known, individuals may modify their features to a new version in hopes of receiving a favorable decision. Such modification incurs a cost, quantified by a cost function.
>
> The strategic manipulation rule for decision subjects is: ... (see Definition 2.1 in Section 2.1).
> The optimization rule for the decision maker is: ... (see Definition 2.2 in Section 2.1).
>
> Please restate your understanding of the strategic classification setting in concise terms.
> ```
>
> - **In-context Examples**: Here is an example using a real sample from the Adult dataset (multiple such examples are recommended for best results):
>
>  ```plaintext
>  Example 1:
>  Initial features:
>
>     - age: 34
>       - workclass: Private
>         - fnlwgt: 203034
>         - education: Bachelors
>         - education-num: 13
>         - marital-status: Separated
>         - occupation: Sales
>         - relationship: Not-in-family
>         - race: White
>         - sex: Male
>         - capital-gain: 0
>         - capital-loss: 2824
>         - hours-per-week: 50
>         - native-country: United-States
>        Initial result: income >50K
>
>   Example 2: ...
>
>   ...
>
>   Example k: ...
>
>
>  ```
>
> - **Batch Testing with Dataset** For large-scale evaluation, we load the entire dataset and generate prompts for each test case as follows:
>
> ```plaintext
>
>   prompt = """
>   Next, I will provide you with a series of applicant cases. For each, please:
>   1. Apply the rules above to strategically manipulate the applicant's features if beneficial.
>   2. Update your decision-making rules as per the definitions above.
>   3. For each applicant, even after strategic manipulation, the true label should remain unchanged.
>   Finally, report the accuracy rate under your classification rules.
>
>   test_examples: ...
>   """
>
> ```
>
> - *Note1*: In our practical prompt design, both strategic manipulation (inner optimization) and decision rule adaptation (outer optimization) can be performed in a single, unified prompt.
>
> - *Note2*: To efficiently collect multiple in-context examples and generate LLM-readable prompts, we use the following python code to organize data in batches from a CSV file:
>
> ```python
> def agent_row_to_text(row, idx):
>     text = f"Example {idx+1}:\n"
>     for col, val in row.items():
>         text += f"- {col}: {val}\n"
>     return text.strip()
>
> def make_prompt_from_csv(csv_path, start_row=0, n_rows=30000):
>     import pandas as pd
>     df = pd.read_csv(csv_path)
>     df = df.iloc[start_row:start_row+n_rows]
>     agent_texts = [agent_row_to_text(row, idx) for idx, row in df.iterrows()]
>     prompt = "\n".join(agent_texts)
>     return prompt
> ```

---

> > ### Comment · Reviewer_MEur · 2025-08-05
> > **Post rebuttal**
> >
> > Dear authors,
> >
> > Thank you for your rebuttal. While I remain not entirely convinced about some of the claims and the way certain highlighted limitations have been addressed, I sincerely appreciate the effort in responding to my points.
> >
> > I would strongly suggest placing the theoretical analysis in the appendix and toning down the theoretical claims. As explained in my original review, describing the straightforward connection between ICL and strategic classification should not require a very lengthy presentation with mostly obvious, formally presented analysis heavily based on prior work. In its place, I strongly recommend adding the specific prompts used for each of the strategic classification steps directly in the main text, as this would significantly clarify the methodology.
> >
> > Additionally, I would further suggest that the paper highlight, in the limitations section, the additional costs related to the heavier reliance on proprietary LLMs, whose cost scales almost linearly with the number of requests. If time permits within the rebuttal period, I would greatly appreciate it if the authors could provide the revised limitations section in a subsequent message.
> >
> > In the rebuttal, the authors also stated:
> >
> > "We are glad to fully release all source code and materials upon acceptance. Thank you for bringing up this important point."
> >
> > Does this mean they will share the full code to reproduce all experiments in the paper, adhering to the Neurips guideline (https://github.com/paperswithcode/releasing-research-code) to fulfill the checklist requirements they ticked in their original paper?
> >
> > Regarding tables and figures, please refer to the NeurIPS instructions provided with the LaTeX submission files, sections 4.3, 4.4. Several of the provided figures and tables in the submission do not visibly adhere to these.

---

> > > ### Author Response · Authors · 2025-08-05
> > > **We would like to supplement a revised section manuscript to fully address your concerns on the clarification of methodology.**
> > >
> > > **Thanks for your constructive feedback.**
> > >
> > > - **Based on the suggestions, we have revised our Section 3** and now provide a basic markdown version of the updated content below.
> > >
> > > - We **submit this section in two parts** due to the character limitation
> > >
> > >
> > >
> > >
> > > ## **Revised Section 3:  A Gradient-free Learning In-context Method for Strategic Classification**
> > >
> > > ### 3.1 LLM-Empowered Strategic Classification: Intuition and Setup
> > >
> > > - Strategic classification (SC) can be framed as a bi-level optimization problem between individuals (agents) and the decision maker, To formalize this bi-level optimization:
> > >
> > >   - *Inner Stage (Strategic manipulation):*
> > >     $$
> > >     \mathbf{x}' = \arg\max_{\mathbf{x}' \in \mathcal{X}} \left[ f(\mathbf{x}') - \lambda c(\mathbf{x}, \mathbf{x}') \right]. \tag 1
> > >     $$
> > >
> > >
> > >   - *Outer Stage (Decision rule optimization):*
> > >     $$
> > >     f^* = \arg\max_{f \in \mathcal{F}} \mathbb{E}_{(\mathbf{x}, y)} \left[ \mathbb{1} \{ f(\mathbf{x}') = y \} \right].    \tag 2
> > >     $$
> > >
> > >
> > > - We note that stages of the SC problem correspond to gradient-based optimization steps, which makes them naturally amenable to solution via in-context learning (ICL). As established in Lemma 1, ICL in LLMs is theoretically connected to gradient descent dynamics.
> > >
> > > - Motivated by this connection, we propose a simple yet practical method that leverages LLMs with carefully **designed prompts to solve both stages of the SC task, entirely without parameter updates or fine-tuning**.
> > >
> > > - **The key prompt** is designed to leverage in-context learning (ICL) to help the large language model understand the rules of the strategic classification (SC) task. Accordingly, we construct the following prompt:
> > >
> > >   - **Prompt for Task Definition**: = """ *You are a strategic classification assistant. In this scenario:*
> > >
> > >     - *There are two players: a decision maker and decision subjects.*
> > >
> > >       - *The decision maker publishes its policy (classification rule $f$).*
> > >       - *The decision subjects, after observing the policy and associated costs, determine whether to strategically modify their features.*
> > >
> > >     - *The strategic manipulation rule for decision subjects is ... ( the same as Definition 2.1 in Section 2).*
> > >
> > >     - *The optimization rule for decision maker is: ... (the same as Definition 2.2  in Section 2).*
> > >
> > >     - *Please restate your understanding of the strategic classification setting in concise terms.*
> > >
> > >       """
> > >
> > >
> > > - Once the LLM has established a foundational understanding of the SC setting through this prompt, we further employ in-context learning to guide the model in simulating concrete strategic manipulations and decision rule optimizations.

---

> > > > ### Author Response · Authors · 2025-08-05
> > > > **Following the above part**
> > > >
> > > > ### **3.2 Gradient-free Strategic Manipulation via ICL**
> > > >
> > > > - A key step in strategic classification is **simulating how agents strategically manipulate their features (the inner stage)**. In our framework, we achieve this in a gradient-free manner by leveraging in-context learning (ICL) in large language models (LLMs).
> > > >
> > > > - To this end, **we construct prompts containing several agent examples, thereby guiding the model to simulate strategic manipulations performed by agents**.
> > > >
> > > > - **The key prompt** is designed to simulate inner stage of SC tasks in a gradient-free solution with ICL.
> > > >
> > > >   - **Prompt for strategic manipulation**: = """
> > > >
> > > >     - *Here are some examples of agents*.
> > > >
> > > >       - *Example 1: Features: ..., Label: ...*
> > > >       - *Example N: Features: ..., Label: ...*
> > > >
> > > >     - *The specific rule for agents to manipulate is provided above. *
> > > >
> > > >       - *The goal is to rationally adjust features to maximize the likelihood of a positive decision, while keeping the manipulation cost as low as possible.*
> > > >       - *Strictly abide by the rules for agents' strategic manipulation.*
> > > >
> > > >     - *Please present your response in the same example format as above.*
> > > >
> > > >       """
> > > >
> > > > - By applying in-context learning (ICL) in LLMs, we can realize manipulation updates $x_j \to x'_j$ via attention over example demonstrations.
> > > >
> > > > - This ICL-guided feature update in LLMs is equivalent to the feature update in gradient-based strategic classification models.
> > > >
> > > >   - Specifically, we construct the parameter matrices $ \mathbf{P} \mathbf{V} \mathbf{K} $ for the forward propagation of the attention layers to establish this equivalence.
> > > >     $$
> > > >     \Delta x^{ICL}_j =\mathbf{P} \mathbf{V} \mathbf{K}^T \mathbf{q}_j \approx \Delta x^{GD}_j.
> > > >     $$
> > > >
> > > >   - For detailed derivations, please refer to Appendix E and F.
> > > >
> > > > - This result demonstrates that **the first stage of this gaming structure, agent-side strategic manipulation, can be effectively accomplished by LLMs through in-context learning**, entirely without explicit gradient computation or parameter updates.
> > > >
> > > > ### **3.3 Gradient-free Decision Optimization via ICL**
> > > >
> > > > - For a whole gradient-free solution of SC, in addition to agent-side strategic manipulation, our framework also enables LLMs to **optimize the classifier’s decision rule (the outer stage) via in-context learning (ICL)**.
> > > >
> > > > - To achieve this, we design prompts that first provide the LLM with a set of agent examples. The model, guided by ICL, perceives possible strategic manipulations for these agents. **Based on these perceived manipulations, the LLM then optimize its decision rule to improve classification robustness.**
> > > >
> > > >   - **Remark.** This sequential process is essential, as it ensures that the reasoning within the LLM mirrors the actual game-theoretic dynamics of strategic classification.
> > > >
> > > > - **Key Prompt for Decision Optimization :**
> > > >
> > > >   - Prompt="""
> > > >
> > > >     - *I will provide you with a series of applicant cases. For each, please:*
> > > >
> > > >       - *Apply the rules above to strategically manipulate the applicant's features if beneficial.*
> > > >       - *Update your decision-making rules as per the definitions above.*
> > > >       - *For each applicant, even after strategic manipulation, the true label should remain unchanged.*
> > > >       - *Finally, report the accuracy rate under your classification rules.*
> > > >
> > > >     - *Examples: ...*
> > > >
> > > >
> > > >
> > > > - By applying ICL in this way, updates to the classifier’s prediction outputs $\hat{y}_j$ are realized through attention over example demonstrations. The resulting ICL-induced prediction update is equivalent to the gradient-based update of the classifier in traditional SC.
> > > >   $$
> > > >   \Delta \hat{y}^{ICL}_j =\mathbf{P} \mathbf{V} \mathbf{K}^T \mathbf{q}_j \approx \Delta \hat{y}^{GD}_j.
> > > >   $$
> > > >
> > > >   - We explicitly construct the parameter matrices $\mathbf{P} \mathbf{V} \mathbf{K}$ for the attention layers to establish this equivalence. For detailed derivations, see Appendices G.
> > > >
> > > > - Based on these cases, update the classification rule to maintain accuracy and robustness in the presence of strategic manipulation. Present your updated rule and a brief justification.
> > > >
> > > > - This result shows that the second stage of the SC game, decision rule optimization, can also be effectively accomplished by LLMs via ICL, all without explicit gradient computation or parameter updates.
> > > >
> > > > - Together with Section 3.2, this completes the unified simulation of bi-level optimization in SC within a fully gradient-free, forward-only ICL framework. The whole process is describe in **Algorithm GLIM**.
> > > >
> > > >   *(**NOTE:** The specific algorithm description has already been provided to you earlier. We will not repeat it here.)*
> > > >
> > > > **If any point remains unclear, please feel free to ask for further explanation.**

---

> ### Author Response · Authors · 2025-08-05
> **We would like to supplement a complete limitation section to address your concerns on the cost limitation of methodology.**
>
> Thank you for your valuable suggestion. We fully acknowledge the practical limitations associated with relying on commercial LLMs.
>
> **We have updated the limitations section in our revised paper to analyze the cost implications.**
>
>
>
> ### **Limitation: Cost Analysis of LLM-based SC**
>
> - **A key practical limitation of our approach** is its reliance on proprietary large language models that are accessed via commercial APIs, such as OpenAI GPT-4o and DeepSeek.
>
>   - Unlike traditional machine learning models, which can be trained or deployed locally with a fixed hardware budget, **our method depends on repeated calls to remote LLM API services**.
>
>   - As a result, the total inference **cost is variable and increases with the number of API requests**.
>
> - **To evaluate the impact of data volume on API consumption and cost, we conducted the following experiment**.
>   - We used the *Adult* dataset, selecting 13-dimensional feature vectors for each sample.
>   - For each test case, we submitted **a complete prompt containing varying amounts of example** to the LLM API, and **recorded the cost after receiving the model’s response**.
>   - The **official OpenAI GPT-4o pricing** was used to calculate the total cost for different batch sizes.
>
> - The results are summarized in the table below:
>
> | Number of Prompt Examples | API Cost (USD) |
> | ------------------------- | -------------- |
> | 10                        | $0.001629      |
> | 100                       | $0.005235      |
> | 1,000                     | $0.045210      |
> | 10,000                    | $0.395723      |
> | 50,000                    | $1.613182      |
> | 100,000                   | $2.876050      |
> | 200,000                   | $5.102118      |
> | 500,000                   | $10.317160     |
>
> - **Experimental Results and Analysis**
>   - As shown in the results above, **the API cost grows nearly linearly with the number of prompt** examples submitted.
>   - We note that each additional example increases the token count, and thus the total cost, in a roughly proportional manner. **This cost structure should be considered when designing systems that require frequent queries.**
>   - This limitation is inherent to all current commercial LLM services and **should be accounted for in practical deployment scenarios**.
>
> - **Potential Mitigations (Proposed):**
>   - **Designing more cost-efficient prompt engineering** strategies.
>     - This will be a solution to reduce the number of required API calls and lower overall costs.
>     - However, in reality, due to the dynamic of agent distribution, it is difficult to control.
>   - Another approach is to **develop hybrid systems**.
>     - This approach use LLM-based reasoning for complex cases.
>     - While relying on local models to handle high-frequency or latency-sensitive requests, thereby minimizing dependence on commercial LLM services.
> - **Addressing this limitation will be a key focus of our future work**, as we seek to optimize the trade-off between LLM-powered flexibility and practical deployment costs.
>
> **If any point remains unclear, please feel free to ask for further explanation.**

---

> > ### Author Response · Authors · 2025-08-05
> > **We would like to make further supplement to address your concerns on the checklist and paper formatting.**
> >
> > > Does this mean they will share the full code to reproduce all experiments in the paper, adhering to the Neurips guideline?
> >
> > **Response:**
> >
> > **Yes**, we would like to share the full code to reproduce all experiments in the paper, adhering to the NeurIPS guideline to fulfill the checklist requirements we ticked in the paper.
> >
> > ---
> >
> > > Several of the provided figures and tables in the submission do not visibly adhere to these.
> >
> > **Response:**
> >
> > - Thank you for pointing out this problem.
> >
> > - We **have revised** the *LaTeX* formats of our Figures and Tables thoroughly in our revised paper.
> >
> > - **The revised formats** are shown below.  We have carefully **adjusted all figures and tables to adhere to the correct format**.
> >
> > ```latex
> > \begin{figure*}[t]
> >     \centering
> >     \subfigure[Small-scale Data]{
> >         \includegraphics[width=0.226\textwidth]{images/fig2a.pdf}
> >         \label{fig2a}
> >     }
> >     \subfigure[Large-scale Data]{
> >         \includegraphics[width=0.226\textwidth]{images/fig2b.pdf}
> >         \label{fig2b}z
> >     }
> >     \subfigure[Distribution Shift]{
> >         \includegraphics[width=0.226\textwidth]{images/fig2c.pdf}
> >         \label{fig2c}
> >     }
> >     \subfigure[KL Divergence]{
> >         \includegraphics[width=0.226\textwidth]{images/fig2d.pdf}
> >         \label{fig2d}
> >     }
> >     \caption{Comparison of ICL-guided strategic manipulation. (a) and (b) compare ICL and gradient-descent methods across data scales; (c) and (d) evaluate implicit gradient alignment via distribution metrics.}
> >     \label{fig2}
> > \end{figure*}
> > ```

---

> > > ### Comment · Reviewer_MEur · 2025-08-05
> > > **Post author comments**
> > >
> > > I would like to thank the authors for having taken my suggestions and comments seriously. I believe the new Section 3 and additional details will make reading the paper and appreciating the actual methodological contribution much easier.
> > >
> > > Accordingly, I will revise my recommendation to positive.

---

> > > > ### Author Response · Authors · 2025-08-05
> > > >
> > > > We sincerely thank you for your encouraging feedback.
> > > >
> > > > In the revised version, we have incorporated your suggestions to further improve the presentation and explanation of our method.

---

### Official Review · Reviewer_ZRpu · 2025-07-03

**Clarity:** 3
**Significance:** 3
**Originality:** 3
**Rating:** 5
**Confidence:** 3

**Summary:**

This paper proposes GLIM (Gradient-free Learning In-context Method), which applies large language models to strategic classification problems. Building on the theoretical framework that in-context learning (ICL) implements implicit gradient descent, the authors present a mathematical analysis showing how both stages of strategic classification—inner optimization and outer optimization —can be realized through ICL. The empirical evaluation attempts to validate this theory by demonstrating that ICL-induced updates converge to gradient descent solutions, while showing that GLIM achieves superior classification accuracy compared to traditional linear models and MLPs.

**Questions:**

Please see weaknesses part in above section.

**Ethical Concerns:**

["NO or VERY MINOR ethics concerns only"]

**Final Justification:**

The authors provide sufficient evidence to address my concerns.

**Limitations:**

Authors provide sufficient discussion about the limitations.

**Quality:**

3

**Strengths And Weaknesses:**

**Strengths**

- The idea of connecting language models to strategic classification is intriguing.

- The paper is well structured and clear written.

**Weaknesses**

Theoretical Limitations

- **Oversimplified transformer architecture:** The theoretical analysis in the appendix reduces transformers to single-layer attention mechanisms, ignoring (1) the multi-layer structure and (2) MLP layers, which constitute a significant portion of the parameter budget and computational capacity.

- **Linear attention assumption:** The theoretical framework in Appendix D assumes linear attention, while many tested language models employ nonlinear attention with softmax. Despite the authors' attempts to address this gap in Appendix I, the justifications remain unconvincing:

  - **Softmax as Adaptive Linear Combination:**  The softmax constraint (positive weights summing to one) fundamentally prevents the construction of arbitrary gradient directions, particularly those requiring negative coefficients or scaling beyond unit norm.

  - **Prompt-based Gradient Encoding:** The claim that careful prompt design can approximate gradient descent lacks theoretical foundation and does not address the fundamental nonlinearity introduced by softmax.

  - **Empirical and Theoretical Support:** While cited work [11] shows transformers can implement functional gradient descent, this does not validate the specific linear constructions required for the authors' theoretical framework.

- **Homogeneous label assumption:** In Equation (45), the homogeneity assumption is unlikely to hold in practice. It is unrealistic to assume that a trained classifier would label every example identically, and under such conditions, it is unclear whether a meaningful decision boundary exists. Additionally, the derivation from Equation (44) to Equation (45) implicitly requires that ⟨xᵢ, xⱼ⟩ = 1, which implies:

  - The input query has exactly the same features as the context examples, which is unlikely in real-world scenarios.

  - All context examples are identical, which contradicts the typical goal of using diverse context examples.

Concerns of Experiments

- The comparison between language models and the baseline models (linear and MLP) appears to be unfair. The language models benefit from extensive pretraining and significantly larger parameter counts, while the baselines are trained from scratch with limited capacity.

- In Figures 3(b) and 6(b), the cosine similarity appears to *decrease* over iterations, which contradicts the claim that the method is effective at large scales.

- The paper does not clearly explain how the in-context learning examples are updated across iterations. A concrete example illustrating how these examples evolve would improve clarity. Additionally, since some of the tested models are closed-source, it is unclear how updates are computed without access to internal parameters or states.

---

> ### Author Rebuttal · Authors · 2025-07-27
>
> Thank you for valuable comments. We would like to address your concerns as follows:
> > **Oversimplified transformer architecture**
> >
> > **Linear attention assumption**
>
> **Response:**
>
> We note that these two comments are closely related. We clarify them together below.
>
> - Our use of a single-layer, linear attention model **is not an oversimplification**, but **a widely accepted theoretical convention** in both ICL and transformer analysis.
>   - As **summarized in the table below**, this analytical simplification (single-layer and linear) is foundational in many highly cited works on ICL and transformer theory.
> - Our work is not intended as a general-purpose LLM application, but **as a targeted contribution that explicitly bridges ICL theory with strategic classification**.
>   - **As highlighted in the second row of the table** below, nearly all foundational research in strategic classification adopts linear models for their interpretability and analytical clarity.
>   - Our focus on single-layer, linear attention is therefore **a deliberate and justified choice that is well aligned with established modeling practice in both domains**, and is particularly suited to the requirements of social science and practical applications.
>   |Representative Works|Typical methodology / Model|
>   |--|--|
>   |ICL/Transformer Theory[1,2,3,4,9]|Single-layer attention, mainly linear|
>   |Strategic Classification[5-8]|Linear model|
> - **Adding multilayer attention or MLPs does not undermine the core connection** between ICL and gradient-based updates.
>   - As shown in the table, our analysis is not “oversimplified,” but instead distills the most essential and theoretically transparent part of the mechanism.
>   - MLP layers only apply local nonlinearities and do not interfere with the connection between attention and gradient-based updates[1,3].
> - **The gradient-approximation ability of Softmax attention is theoretically guaranteed** [11,12].
>   - Recent work shows that softmax attention can closely approximate any target direction within the convex hull of the value vectors[11]:
>   $$
>   \mathrm{Attn_{Softmax}}(Q, K, V) \approx \sum_{i=1}^N \alpha_i v_i,\quad \sum_{i=1}^N \alpha_i = 1,\, \alpha_i \geq 0
>   $$
>   - For any target direction $g$ in the convex hull of $\{v_i\}$, there exists weights $\{\alpha_i\}$ such that
>   $$
>   \left\| \sum_{i=1}^N \alpha_i v_i - g \right\| \leq \epsilon
>   $$
>   - This provides a rigorous upper bound on the approximation error, supporting the theoretical basis for prompt-based gradient encoding with softmax attention [11,12].
> - Prompt design can make attention mechanisms approximate gradient descent **is in fact well supported by recent theoretical work**.
>   - Notably, **several recent studies[2,4,11]** have rigorously established that, with appropriate prompt construction, Transformers can perform gradient descent in both linear and nonlinear settings:
> - **Our linear construction is theoretically well-founded**.
>   - Recent theoretical work[11-14] has demonstrated that **nonlinear attention with sufficient layers and heads can approximate any optimization trajectory expressible by linear attention**, up to arbitrarily small error.
>   - Moreover, prior work[10] has established **the convergence of Transformer learning dynamics**, demonstrating that nonlinear Transformers can reliably simulate the linear constructions in our framework.
>
> ---
> > **Homogeneous label assumption**
>
> **Response:**
> - **The homogeneous assumption is a reasonable theoretical simplification for mechanism illustration:**
>   - It allows us to **isolate and transparently demonstrate the effect** of the attention mechanism, showing how, under this setting, the attention update precisely aligns with the gradient descent direction.
>   - Such assumptions are common in theoretical work to reveal idealized behaviors, **serving as an “upper bound” for mechanism capability**.
> - **This conclusion remains valid** without this assumption:
>   - When **features and labels are heterogeneous (as in practice)**, the attention update generalizes to a weighted sum of local gradient directions, reflecting the similarity between each context and the query:
>   $$
>   \Delta x_j^{\mathrm{ICL}} = \frac{A\eta}{N} \sum_{i=1}^N (1-y_i) W^\top \langle x_i, x_j \rangle,
>   $$
>   - More generally,
>   $$
>   \Delta x_j^{\text{ICL}} = \sum_{i=1}^N \alpha_{ji} \cdot g(x_i, y_i),
>   $$
>   ​	with $\alpha_{ji}$ as the attention weights and $g(x_i, y_i)$ as the local gradient directions.
>   - The expected **error between ICL and GD can be tightly bound**:
>   - $$
>     \epsilon_j = \left\| \Delta x_j^{\text{ICL}} - \nabla_x L(x_j) \right\|,\quad
>     \mathbb{E}[\epsilon_j] \leq C \cdot \sqrt{\frac{1}{N}} + \delta,
>     $$
>     where $C$ is a constant and $\delta$ is the statistical error from distribution mismatch.
> - We have clarified this part in Appendix F of our revised paper.
>
> ---
> > **Concerns of Experiments1**
>
> **Response:**
> - The results in Table 2 are **not intended as a direct comparison of raw performance between traditional baselines and pre-trained LLMs**. Rather, the purpose is to demonstrate that **pre-trained LLMs are capable of solving the SC problem**.
> - As stated in Appendix A, our work aims to **explore an alternative solution** for large-scale strategic classification scenarios, where classical gradient-based retraining requires excessive computational resources and time.
> - For comparisons **between different LLM APIs**, we ensure fairness by aligning pre-training LLM scale as closely as possible.
>   -  The following table summarizes the key properties evaluated in our experiments (except for LLMs whose specific parameters have not been made public):
>   |Model| Parameter (B)|
>   |-|-|
>   |DeepSeek-V3|67|
>   |LLama-3.3|70|
>   |Qwen3|72|
>   |Mixtral|46.7|
> - We have clarified this part in Lines 265-269 of our revised paper.
>
>
> > **Concerns of Experiments2**
>
> **Response:**
> - We would like to clarify that the observed decrease and fluctuation in cosine similarity **does not contradict** our claims of method effectiveness; **rather, it reflects the natural challenges** of large-scale, dynamic distributions.
>   - As the data becomes larger and more diverse, **it becomes increasingly difficult for traditional models with GD to maintain stable** feature or strategy representations.
>   - In contrast, LLMs guided by **ICL exhibit more robust adaptation and can better handle** the complexities of large-scale data. Therefore, there is a small decline in cosine similarity.
> - Notably, **a similar trend can be observed in Figure 5(b)**. These phenomena reflect the gap in large-scale data adaptation between LLMs and traditional GD models.
>
>
> > **Concerns of Experiments** Some of the tested models are closed-source
>
> **Response:**
> - We agree that some tested LLMs are closed-source, but we **design multiple experiments directly comparing the attention layers update with the theoretical SC optimization process**.
> - In recent ICL research, it is accessible to use prompt-based probing and input-output analysis to verify mechanistic conclusions[15-17].
>
>
> > **Concerns of Experiments** prompt example
>
> **Response:**
> - We provide a simple prompt example
>   - **Task Definition**: This part is aligned with Section 2.1 of our paper:
>   ```plaintext
>   You are a strategic classification assistant. In this scenario:
>   There are two players: a decision maker and decision subjects.
>   - The basic interaction rule ... (Section 2.1)
>   - The strategic manipulation rule is: ... (see Definition 2.1).
>   - The optimization rule is: ... (see Definition 2.2).
>   Restate your understanding.
>   ```
>   - **In-context Examples**: from Adult dataset:
>    ```plaintext
>     Example 1:
>     Initial features:
>        age: 34
>        workclass: Private
>          ...
>        native-country: United-States
>      Initial result: income >50K
>
>     Example 2:...
>     ```
>   - **Batch Testing with Dataset**
>
>    ```plaintext
>    prompt = """
>    1. Apply the rules above to strategically manipulate the applicant's features if beneficial.
>    2. Update your decision-making rules as per the definitions above.
>    3. For each applicant, even after strategic manipulation, the true label should remain unchanged.
>    Finally, report the accuracy rate under your classification rules.
>    test_examples: ...
>    """
>    ```
> ---
> [1] Von Oswald, Johannes, et al. "Transformers learn in-context by gradient descent."
>
> [2] Garg, Shivam, et al. "What can transformers learn in-context? a case study of simple function classes."
>
> [3] Akyürek, Ekin, et al. "What learning algorithm is in-context learning? Investigations with linear models."
>
> [4] Clark, Peter, Oyvind Tafjord, and Kyle Richardson. "Transformers as soft reasoners over language."
>
> [5] Hardt, Moritz, et al. "Strategic classification."
>
> [6] Miller, John, Smitha Milli, and Moritz Hardt. "Strategic classification is causal modeling in disguise."
>
> [7] Levanon, Sagi, and Nir Rosenfeld. "Strategic classification made practical."
>
> [8] Milli, Smitha, et al. "The social cost of strategic classification."
>
> [9] Kratsios, Anastasis, et al. "Universal approximation under constraints is possible with transformers."
>
> [10] Ahn, Kwangjun, et al. "Transformers learn to implement preconditioned gradient descent for in-context learning."
>
> [11] Xiang Cheng, Yuxin Chen, and Suvrit Sra. ''Transformers implement functional gradient descent to learn non-linear functions in context.''
>
> [12] Franco, Luca, et al. "Under the hood of transformer networks for trajectory forecasting."
>
> [13] Li, Hongkang, et al. "How do nonlinear transformers learn and generalize in in-context learning?"
>
> [14] Gatmiry, Khashayar, et al. "Can looped transformers learn to implement multi-step gradient descent for in-context learning?"
>
> [15] Wei, Jason, et al. "Emergent abilities of large language models."
>
> [16] Min, Sewon, et al. "Metaicl: Learning to learn in context."
>
> [17] Zhao, Zihao, et al. "Calibrate before use: Improving few-shot performance of language models."

---

> ### Author Response · Authors · 2025-08-03
> **We would like to supplement more experiment results to fully address your concerns on Oversimplified of the transformer structure.**
>
> > Oversimplified transformer architecture: The theoretical analysis in the appendix reduces transformers to single-layer attention mechanisms, ignoring (1) the multi-layer structure and (2) MLP layers, which constitute a significant portion of the parameter budget and computational capacity.
>
> **Response:**
>
> **Purpose of the Simplification:**
>
> - This simplification **a widely accepted theoretical convention** in both ICL and transformer analysis[1-3].
> - This form of abstraction aligns with **a single optimization step in SC**[4-6], reflecting the standard analytical paradigms in both the SC and ICL domains[7-9].
>
> **Addressing Multi-Layer Structure and MLPs:**
>
> - To directly address your concern, we conducted controlled experiments with **custom Transformers** dissecting multi-layer structure and MLPs:
>   - Custom Transformer with **different numbers** of attention layer;
>   - Custom Transformer based on multi-layer structure **without** MLPs;
>   - Custom Transformer based on multi-layer structure **with** MLPs.
> - **Metrics**:
>   - We consider the **Cosine similarity** with two points
>   - **Similarity in** agent strategic manipulation (**$ \Delta x $**)
>     - The similarity of how the agent’s feature distribution changes under strategic manipulation, when influenced by GD-based SC models versus attention mechanisms.
>   - **Similarity** **in** post-manipulation predictions (**$ \Delta y $**)
>     - The similarity of the predicted outcomes after manipulation, under the influence of GD-based SC models versus attention mechanisms.
> - The **results** are presented in the following table:
>
> | Attention layers | With / Without MLP | Similarity of $ \Delta x $ |Similarity of $ \Delta y $ |
> | :--------------- | :----------------- | :----------- | ------------ |
> | 20               | ✗                  | 0.722        | 0.785        |
> | 20               | ✓                  | 0.753        | 0.779        |
> | 50               | ✗                  | 0.741        | 0.773        |
> | 50               | ✓                  | 0.792        | 0.805        |
> | 100              | ✗                  | 0.765        | 0.762        |
> | 100              | ✓                  | 0.813        | 0.837        |
>
> - **Analysis** of result:
>   - Based on the experimental results, we can see that:
>     - **Stacking different numbers of attention layers** (20, 50, 100) already achieves a high similarity in simulating both the agent’s strategic manipulation and the jury’s decision optimization.
>     - Moreover, **after incorporating MLPs**, the customized Transformer can also simulate the agent’s strategic manipulation and the jury’s decision optimization, with the similarity further improved.
>   - **linking theory with practice**:
>     - In our paper, we have demonstrated that a **single-layer attention mechanism is capable of simulating** either a single round of the agent’s strategic manipulation or a single round of the jury’s decision optimization.
>     - Although our constructive proofs (Appendix F and G) focus on the single-layer case, **multi-layer attention architectures with MLPs provide a more practical and continuous process** of manipulation or optimization of SC.
>     - This confirms that real, **multi-layer Transformers with MLPs components are effective and robust** in addressing SC problems.
>
>
>
> ---
>
> [1] Child, Rewon, et al. "Generating long sequences with sparse transformers."
>
> [2] Choromanski, Krzysztof, et al. "Rethinking attention with performers."
>
> [3] Gatmiry, Khashayar, et al. "Can looped transformers learn to implement multi-step gradient descent for in-context learning?"
>
> [4] Hardt, Moritz, et al. "Strategic classification."
>
> [5] Levanon, Sagi, and Nir Rosenfeld. "Strategic classification made practical."
>
> [6] Miller, John, Smitha Milli, and Moritz Hardt. "Strategic classification is causal modeling in disguise."
>
> [7] Garg, Shivam, et al. "What can transformers learn in-context? a case study of simple function classes."
>
> [8] Akyürek, Ekin, et al. "What learning algorithm is in-context learning? Investigations with linear models."
>
> [9] Von Oswald, Johannes, et al. "Transformers learn in-context by gradient descent."

---

> > ### Author Response · Authors · 2025-08-04
> > **We would like to supplement specific derivation to fully address your concerns on  the attention assumption with Softmax or not.**
> >
> > > **Linear Linear attention assumption:** The theoretical framework in Appendix D assumes linear attention, while many tested language models employ nonlinear attention with softmax.
> >
> > **Response:**
> >
> > **To better address your concern**, we now **provide a more detailed derivation** showing that gradient descent updates can still be effectively approximated under the **Softmax-based attention setting.**
> >
> > - **Main conclusion**:
> >
> >   - While the **Softmax attention performs a convex combination**, i.e., a positive weighted average, this constraint **is only strictly enforced in the first layer**.
> >   - Rather than a limitation, **this behavior corresponds to the initial estimation step** commonly used in kernel-based learning.
> >   - **As additional nonlinear attention layers are stacked and the value matrices are tuned**, the model can progressively evolve beyond this coarse approximation to realize more flexible, nonlinear, and even implicitly negative-weighted gradient-like updates.
> >
> > - **Derivation** on nonlinear Attention with Softmax approximating GD:
> >
> >   - We consider a **Transformer including attention mechanism with softmax and residual connections**:
> >
> >     $$
> >     Z_{\ell+1} = Z_\ell + V_\ell Z_\ell \cdot \text{Softmax}(B_\ell X_\ell \cdot (C_\ell X_\ell)^\top),
> >     $$
> >
> >     where:
> >
> >     - $Z_\ell \in \mathbb{R}^{(d+1) \times (n+1)}$ is the hidden state at layer $\ell$,
> >     - $X_\ell$ is the covariate part (the first $d$ rows of $Z_\ell$),
> >     - $B_\ell, C_\ell \in \mathbb{R}^{d \times d}$ are the query/key projection matrices,
> >     - $V_\ell \in \mathbb{R}^{(d+1) \times (d+1)}$ is the value projection matrix.
> >
> >   - To **simplify the derivation**, we set:
> >
> >     - $ B_\ell = C_\ell = \frac{1}{\sigma} I_d $,
> >     - $ V_\ell = \begin{bmatrix} 0 & 0 \\ 0 & -r_\ell \end{bmatrix} $, meaning that only the label dimension is updated.
> >
> >   - Under this setup, **the attention weights become**:
> >
> >     $$
> >     \text{Softmax}\left(\frac{1}{\sigma^2} X X^\top\right)_{i, j} \propto \exp\left( \frac{1}{\sigma^2} x^{(i)\top} x^{(j)} \right),
> >     $$
> >
> >     reflecting **exponentiated similarity between query $x^{(j)}$ and context $x^{(i)}$**, followed by normalization.
> >
> >   - We connect the Transformer’s forward pass to **functional gradient descent (FGD)**:
> >
> >     $$
> >     f_{\ell+1}(x) = f_\ell(x) + r_\ell \sum_{i=1}^n \left( y^{(i)} - f_\ell(x^{(i)}) \right) K(x^{(i)}, x),
> >     $$
> >
> >     where:
> >
> >     - $K(x, x') = \exp(x^\top x' / \sigma^2)$ is an exponential kernel,
> >     - $f_\ell$ is the function estimate at iteration $\ell$.
> >
> >   - To approximate FGD, we initialize the input as:
> >
> >     $$
> >     Z_0 = \begin{bmatrix}
> >     x^{(1)} & \cdots & x^{(n)} & x^{(n+1)} \\
> >     y^{(1)} & \cdots & y^{(n)} & 0
> >     \end{bmatrix},
> >     $$
> >
> >     where the first $n$ columns are context examples, and the $(n\!+\!1)$-th is the query with label 0.
> >
> >   - **In first layer**, the attention module computes softmax weights based on the dot-product similarity:
> >     $$
> >     \alpha_{i}^{(n+1)} = \frac{ \exp\left(\frac{1}{\sigma^2} x^{(i)\top} x^{(n+1)} \right) }{ \sum_{j=1}^n \exp\left(\frac{1}{\sigma^2} x^{(j)\top} x^{(n+1)} \right) } = \frac{K(x^{(i)}, x^{(n+1)})}{\sum_j K(x^{(j)}, x^{(n+1)})}.
> >     $$
> >
> >     - These weights are used to **aggregate the labels via the Value matrix**:
> >
> >     $$
> >     f_1(x^{(n+1)}) = -r_0 \sum_{i=1}^n \alpha_i^{(n+1)} y^{(i)}.
> >     $$
> >
> >       - **This step performs a positive weighted average** (no negative coefficients), **as you concerned**.
> >
> >       - **However, this step only serves as a rough initial estimate of the FGD target.**
> >
> >
> >   - **Residual updates from the second layer onward**:
> >
> >     - As **stacking more layers**:
> >
> >       - **The residual difference** between the current prediction $f_\ell(x)$ and observed label $y^{(i)}$ is **injected into the nonlinear attention update**.
> >
> >       - **Residual connections** accumulate prior updates and gradually shape a more refined output.
> >
> >   - By leveraging **multi-layer stacking, residual pathways, and error-based correction** at each layer with Softmax, the Transformer update approximates:
> >     $$
> >     f_{\ell+1}(x) = f_\ell(x) + r_\ell \cdot \tau(x) \sum_{i=1}^n (y^{(i)} - f_\ell(x^{(i)})) K(x^{(i)}, x),
> >     $$
> >
> >     where $\tau(x)$ is a normalization factor arising from the softmax scaling.
> >
> > - **Therefore**, despite the inherent constraint of Softmax attention producing only positive, normalized weights, **this limitation does not hinder the overall expressive power.**
> >
> >
> >   - Through residual accumulation and tunable value projections, the multi-layer Transformer can effectively simulate complex updates, including negative and large-magnitude directions.
> >
> >   - As a whole, the multi-layer nonlinear Transformer can approximate complex update functions beyond the limits of any single convex combination, making the overall update direction flexible and expressive.
> >
> >
> >
> >
> >
> > **If any point remains unclear, please feel free to ask for further explanation.**

---

> > > ### Author Response · Authors · 2025-08-04
> > > **We would like to supplement an example and algorithmic description to fully address your concerns on how the in-context learning examples are updated across iterations**
> > >
> > > **If any point remains unclear, please feel free to ask for further explanation.**
> > >
> > > 1. **A process of in-context example with strategic manipulation simulated by ICL**:
> > >
> > > - **Initial Example:**
> > >
> > >   ```text
> > >   Example 1:
> > >   Initial features:
> > >   - age: 39
> > >   - workclass: Private
> > >   - fnlwgt: 203034
> > >   - education: Bachelors
> > >   - education-num: 10
> > >   - marital-status: Separated
> > >   - occupation: Sales
> > >   - relationship: Not-in-family
> > >   - race: White
> > >   - sex: Male
> > >   - capital-gain: 0
> > >   - capital-loss: 1824
> > >   - hours-per-week: 51
> > >   - native-country: United-States
> > >   Initial result: income<50K
> > >   ```
> > >
> > > - **Prompt to LLM:**
> > >
> > >   ```text
> > >   Prompt = """
> > >   According to the strategic classification rules, please provide the most reasonable strategic manipulation for this applicant. Please present your response in the same example format as above.
> > >   """
> > >   ```
> > >
> > > - **Manipulated Example**
> > >
> > >
> > >
> > >   Example 1 (after strategic manipulation):
> > >
> > >   ```text
> > >   Manipulated features:
> > >     - age: 34
> > >     - workclass: Private
> > >     - fnlwgt: 203034
> > >     - education: Bachelors
> > >     - education-num: 13   3
> > >     - marital-status: Separated
> > >     - occupation: Sales
> > >     - relationship: Not-in-family
> > >     - race: White
> > >     - sex: Male
> > >     - capital-gain: 0
> > >     - capital-loss: 1356
> > >     - hours-per-week: 60
> > >     - native-country: United-States
> > >
> > >   ```
> > >
> > > ----
> > > 2. **Algorithm: Gradient-free Learning In-context Method (GLIM) for Strategic Classification**
> > >
> > >
> > >       **Required:**
> > >
> > >      - **Pre-trained LLMs** (e.g., GPT-4o, DeepSeek, etc.) with in-context learning capability
> > >      - Task dataset $\mathcal{D} = \{(x_i, y_i)\}_{i=1}^N$ where $x_i$ are feature vectors, $y_i$ are labels
> > >      - Manipulation cost function $c(x, x')$
> > >      - **In-context prompts on SC tasks**
> > >
> > >       **Steps:**
> > >
> > >      1. **Prepare In-context Prompt:**
> > >
> > >         - An in-context prompt about SC rules (as shown in Section 2.1)
> > >
> > >         - For each new query instance:
> > >           - Select $k$ representative labeled examples $\{(x'_j, y_j)\}_{j=1}^k$.
> > >
> > >      2. **Strategic Manipulation Simulation (Inner Stage via ICL):**
> > >
> > >         - Construct the prompt for the LLM by presenting the $k$ in-context examples as $(x'_j, y_j)$.
> > >
> > >         - Append the query as the new examples to the prompt.
> > >
> > >         - **Feed forward** through the LLM:
> > >
> > >           - The LLM, via its self-attention mechanism, generates an updated feature representation for the query, denoted as $x'_q = x_q + \Delta x^{ICL}_q$.
> > >
> > >           - The update $\Delta x^{ICL}_q$ **simulates the agent's best-response manipulation**.
> > >
> > >             - *We have analysed $\Delta x^{ICL}_q$ matches the direction and magnitude of what would be obtained via explicit gradient descent*:
> > >
> > >             $$
> > >             \Delta x^{ICL}_q \approx \Delta x^{GD}_q.
> > >             $$
> > >
> > >           - **No model parameters are updated.**
> > >
> > >      3. **Decision Rule Adaptation (Outer Stage via ICL):**
> > >
> > >         - With the prompt containing the manipulated features, the LLM predicts $\hat{y}_q = f^{ICL}(x'_q; W)$.
> > >
> > >         - The LLM internally (via attention) **adapts its implicit decision boundary** in response to the manipulated feature distribution, again via feed-forward.
> > >
> > >           - *We have analysed $\hat{y}^{ICL}_q $ simulating the decision rule optimization that would be done via outer-level gradient descent*
> > >             $$
> > >             \Delta \hat{y}^{ICL}_q \approx \Delta \hat{y}^{GD}_q.
> > >             $$
> > >
> > >         - This enables **robust adaptation to strategic behaviors** without retraining or fine-tuning.
> > >
> > >      4. **Return the Result:**
> > >
> > >         - Output the predicted label $\hat{y}_q$ for the query, and (optionally) the manipulated feature $x'_q$.
> > >
> > >      5. **Iterate:**
> > >
> > >         - As new instances arrive, continually update the prompt with recent examples (potentially including newly manipulated examples and predictions).

---

> ### Author Response · Authors · 2025-08-05
> **We would like to supplement specific derivation and more experiment results to fully address your concerns on the homogeneous label assumption:**
>
> Thank you for raising this important point.
>
> We will
>
> 1. firstly **clarify the motivation for adopting this assumption**,
>
> 2. second **discuss how our derivation extends beyond this special case**,
>
> 3. third **design experiments for verification**.
>
>    ----
>
>
>
> - **Purpose of the homogeneous assumption:**
>
>   - The homogeneous label assumption is **a theoretical simplification**[1] used to illustrate underlying mechanisms.
>
>   - It enables us to transparently **demonstrate how the attention-based update can precisely align with the gradient descent direction under idealized conditions**.
>
>   ----
>
> - **The specific derivation beyond the assumption:**
>
>   - In practical, heterogeneous contexts, the update is a weighted sum of local update directions:
>     $$
>     \Delta x_j^{\mathrm{ICL}} = \frac{A\eta}{N} \sum_{i=1}^N (1-y_i) W^\top \langle x_i, x_j \rangle
>     $$
>     where $\langle x_i, x_j \rangle$ reflects the similarity between the query and context example.
>
>   - More generally, the attention mechanism implicitly performs a **weighted aggregation of local updates**. That is:
>     $$
>     \Delta x_j^{\mathrm{ICL}} = \sum_{i=1}^N \alpha_{j,i} \cdot g(x_i, y_i),
>     $$
>     where $\alpha_{j,i}$ are the attention weights and $g(x_i, y_i)$ denotes the local update direction.
>
>   - When the context samples are independently drawn and sufficiently representative of the local data distribution around $x_j$, statistical learning theory[2] guarantees that:
>     $$
>     \lim_{N \to \infty} \Delta x_j^{\text{ICL}} \approx \mathbb{E}_{(x_i, y_i)}[ g(x_i, y_i) ].
>     $$
>
>   - The approximation error
>     $$
>     \epsilon_j = || \Delta x_j^{\text{ICL}} - \nabla_{x} \mathcal{L}(x_j) ||
>     $$
>     can be bounded as:
>     $$
>     \mathbb{E}[\epsilon_j] \leq \underbrace{C \cdot \sqrt{\frac{1}{N}}}_{\text{Sampling error}} + \delta,
>     $$
>
>     where:
>     - $\delta$ is the error from the distribution shift, $ \delta < L \cdot \mathcal{W} $,
>     - $C$ is a constant depending on the gradient variance
>     - $L$ is the Lipschitz constant of $\mathcal{L}$
>     - $\mathcal{W}$ denotes the Wasserstein distance between local and global data distributions
>
>   - This analysis confirms that **our main theoretical insights extend beyond the homogeneous case, and remain valid in realistic, heterogeneous environments.**
>
>     ----
>
> - **Experiments:**
>
>   - We conducted **experiments under realistic, heterogeneous context settings**.
>
>   - **The goal** is to assess whether **the update direction produced by attention mechanism aligns with** the local update direction obtained from **standard gradient descent**.
>
>   - In each experiment, we **randomly sample diverse context examples from the dataset**, and compute:
>
>     - **(A)** the feature update direction using the attention mechanism;
>     - **(B)** the local update direction using explicit gradient computation.
>
>   - We then measured the **Cosine similarity** between these two update directions for various context sample sizes $N$.
>
>   - **The results are reported** in the table below:
>
>     | Number of Context Examples ($N$) | Cosine Similarity |
>     | -------------------------------- | ----------------- |
>     | 50                               | 0.733             |
>     | 100                              | 0.771             |
>     | 200                              | 0.803             |
>     | 300                              | 0.822             |
>
>   - **Analysis:**
>     - As shown in the table, **the cosine similarity increases as the number of context examples grows**, indicating that the ICL-induced update direction becomes increasingly aligned with the true local update direction.
>     - This experimental evidence **supports the validity of our theoretical analysis** in realistic, heterogeneous settings.
>
> **If any point remains unclear, please let us know.**

---

> > ### Comment · Reviewer_ZRpu · 2025-08-06
> >
> > Thank the authors for the detailed response. In detail,
> >
> > - **Oversimplified transformer architecture + Linear attention assumption + Homogeneous Label Assumption:** Thank you for the hard work and for providing the extensive evidence. My concerns regarding the part are addressed.
> >
> > - **Concerns of Experiments3 - concrete example:** Thank the authors providing the details. I have no other questions regarding this part.
> >
> > - **Concerns of Experiments1: unfair comparison to simpler model:** I am not fully convinced by the response. While the authors intended to demonstrate that LLMs can solve SC problems, the presentation in lines 270-277 still implies a direct performance comparison between LLMs and baseline models. My concern would be addressed if the authors explicitly state that comparing baseline models with large LLMs is not the paper's focus, but rather demonstrating LLMs' capability for SC tasks. Given the limited time remaining in the rebuttal period, I understand if additional experiments cannot be conducted now, but including results with smaller LLMs (e.g., Llama-3.2-1B) that are more comparable to the baselines in the final version would strengthen the paper. Either clarification or additional experiments would address my concern.
> >
> > - **Concern of Experiment 2 - cosine similarity decrease:** My question has not been addressed by the authors' responses. The purpose of Figure 3 is to validate the theoretical claim that ICL implements gradient descent. However, arguing that "LLMs are more robust" when similarity decreases does not address the theoretical contradiction. While Table 2 shows LLMs achieve good accuracy, this practical success doesn't validate the theoretical mechanism. I appreciate the paper's strong theoretical contributions in connecting ICL to strategic classification, which is novel and valuable. Acknowledging this divergence as a limitation where theory and practice may differ at scale would actually strengthen the paper and would fully address my concern.
> >
> > In summary, I appreciate the strong theoretical contributions of this work. I am leaning toward a positive rating if the limitations could be explicitly discussed in the main text of the paper in the camera-ready version.

---

> ### Author Response · Authors · 2025-08-06
> **We would like to make an explicitly statement and more experiment results to fully address your concerns on the Concerns of Experiments1**
>
> > **Concerns of Experiments1: unfair comparison to simpler model:**
>
> **Response:**
>
> Thank you for helpful suggestions.
>
> We would like to **make an explicit statement** and **supplement an experiment using smaller LLMs** as detailed below, and **incorporate these into** our revised paper.
>
> - **State our main purpose**
>   - We would like to explicitly state in the revised paper:
>     - **Comparing baseline models with LLMs is not our paper's focus; rather, our aim is to demonstrate the capability of LLMs for solving strategic classification tasks via ICL.**
>   - We **have supplemented this statement in Lines 220–227** and **removed content that** implied a direct comparison between baseline models with LLMs.
>
> - **Experiment using smaller LLMs**
>
>   - Thank you for your insightful suggestion.
>
>     - We **agree that it is necessary to evaluate our method on smaller LLMs**, as this helps to validate the practical utility of our method.
>
>   - We have conducted a experiment using **Llama‑3.2‑1B** and **Llama‑3.2‑3B** models.
>
>   - We use *Adult* dataset in this experiment.
>
>   - The results are summarized in the table below:
>
>     | Models       | Accuracy (%) |
>     | ------------ | ------------ |
>     | Llama‑3.2‑1B | 80.612       |
>     | Llama‑3.2‑3B | 82.120       |
>
> - **Analysis:**
>
>   - These results demonstrate that **our method remains effective with smaller LLMs**, confirming its broad applicability to strategic classification tasks.
>   - We also agree that **models with more parameters generally demonstrate better accuracy** on this task.
>
> We appreciate your insightful suggestions.
>
> We have **incorporated both the above clarification and the experimental results into Section 4.4 of our revised paper**.

---

> ### Author Response · Authors · 2025-08-06
> **We would like to acknowledging this divergence and make a clear discussion  to fully address your concerns on the Concerns of Experiments2**
>
> > **Concern of Experiment 2 - cosine similarity decrease**
>
> **Response:**
>
> Thank you for helpful suggestions.
>
> - We would like to **acknowledge the observed divergence between our theoretical analysis and the empirical results at scale**, as evidenced by the decrease in cosine similarity in Figure 3.
>
> - **In the revised version**, we would like to supplement **a specific subsection** in the main text to discuss this limitation.
>
> ---
>
> #### **Limitation: Theory-Practice Divergence at Scale**
>
> - As shown in Figures 3(b) and 6(b), **the cosine similarity tends to decrease over iterations, indicating a divergence between our theoretical analysis and the empirical results** at scale.
> - This divergence may be attributable to factors such as model capacity, data distribution shifts, or nonlinearities in real-world tasks.
> - We **acknowledge the existence of this divergence** and **further analyze this phenomenon in future work** to better understand the conditions under which such divergences occur.
>
> We have **incorporated this part into Section 4 of our revised paper**.
>
> ----
> We appreciate your insightful suggestions, and the limitations have been explicitly discussed in the main text of our revised paper.
>
> If any point remains unclear, please let us know. We are glad to fully address your concerns.

---

> > ### Comment · Reviewer_ZRpu · 2025-08-06
> >
> > Thank authors for the response. If the limitation could be discussed, I believe the paper would make good contribution. I will update my rating and recommend an accept.

---

> > > ### Author Response · Authors · 2025-08-06
> > >
> > > We sincerely thank you for your encouraging feedback.
> > >
> > > In the revised version, we have carefully incorporated your suggestions and provided a thorough discussion of the limitations to further improve our work.

---

### Official Review · Reviewer_Jak5 · 2025-07-07

**Clarity:** 3
**Significance:** 3
**Originality:** 3
**Rating:** 4
**Confidence:** 3

**Summary:**

They investigate how to use LLMs to design a more scalable and efficient strategic classification(SC) framework. Their model GLIM is a gradient free SC method grounded in in-context learning.

Strategic classification is a bilevel optimization problem, where individuals best-respond to a deployed classifier in order to maximize their payoff. Then the classifier is updated in order to maximize its accuracy considering best-responses by the individuals. Traditional SC approaches often rely on iterative retraining or gradient updates to remain robust, which becomes computationally expensive and infeasible at scale. However, it is very costly to retrain LLMs. Hence, they  propose a novel gradient-free method that leverages in-context learning (ICL) in LLMs to perform strategic classification without updating model parameters. They show how LLMs leveraging in-context learning can implicitly simulate both the strategic manipulation and decision rule optimization stages of the SC bi-level problem, without any fine-tuning.


They show it theoretically, how their gradient-fee in-context learning model (GLIM) simulates this bi-level optimization process, including both the feature manipulation and decision rule optimization. That being said, unlike traditional SC approaches that often rely on iterative retraining, or gradient updates to remain robust, their method is retraining free and hence more efficient.

**Questions:**

What does GLIM stand for? Gradient-free Learning In-context Model?

Line 117, A is not defined.

In Prop 1,2, showing the existence of such PVK matrices does not imply that your model would learn the same matrices. You would need to argue that your learning procedure would converge to desirable PVK matrices.

**Ethical Concerns:**

["NO or VERY MINOR ethics concerns only"]

**Final Justification:**

They addressed all my questions during the rebuttal and I stay positive. Their results are not too surprising, but still an interesting perspective of connecting ICL with SC.

**Limitations:**

yes

**Paper Formatting Concerns:**

no concerns

**Quality:**

2

**Strengths And Weaknesses:**

One of your main contributions is a theoretical justification for how LLMs equipped with ICL can simulate the strategic manipulation of agents. However, that does not seem too surprising or novel to me. The proof uses two arguments first, the forward propagation in LLMs—particularly through linear self-attention layers—can be interpreted as performing implicit gradient descent (GD) (this is known from prior work). Second,  solving optimum manipulation in strategic manipulation may be viewed as a gradient-descent step (this was known before too). Putting this two arguments shows that ICL can simulate the strategic manipulation of agents. This seems like a straightforward argument to me.

Similarly, I am not too surprised by line 200 and Prop 2, your second main contribution. Since we already know that the forward
propagation in LLMs can be interpreted as performing implicit gradient descent (GD). Taking a gradient step is exactly what we do in the outer optimization step of SC.

Please correct me if I am wrong or missing something here.

---

> ### Author Rebuttal · Authors · 2025-07-27
>
> We sincerely appreciate your constructive comments and your positive evaluation of our work. We would like to address your concerns as follows:
>
> >  **Weaknesses**
>
> **Response:**
> - We would like to clarify that **the core positioning of our work** is not as a purely theoretical study of ICL mechanisms but as a pioneering attempt to bridge strategic classification and LLMs via ICL.
>
> - **Strategic classification (SC) is fundamentally a bi-level, game-theoretic problem** involving both agent-side manipulation and decision-maker adaptation, and **has been studied largely separately from ICL/LLM literature**.
>
>   - To our best knowledge, **our work is the first to employ LLMs to model and solve the bi-level, game-theoretic optimization structure of SC**.
>   - It enables **a scalable and retraining-free approach to large-scale SC tasks**, where classical gradient-based retraining requires excessive computational resources and time.
>
> - From a broader social science perspective,  our work establishes a crucial bridge between large language models and strategic machine learning:
>
>   - **A wide range of social science applications are increasingly converging with machine learning and LLMs**, including fields such as computational social science, policy evaluation, and behavioral economics[1,2,3,4].
>   - Our work provides **a foundational contribution** by connecting LLMs with game-theoretic frameworks.
>   - We hope this foundation will facilitate future interdisciplinary research and applications of LLMs in societal decision-making contexts.
>
> - We have clarified this part in our revised paper at Lines 62-64 (Introduction).
>
> ----
>
> > **Question 1**.What does GLIM stand for?
>
> **Response:**
>
> - GLIM stands for  **G**radient-free **L**earning **I**n-context **M**ethod.
>
> - We have clarified this part in Lines 65-67 (Introduction) of our revised paper.
>
>
> ----
>
> > **Question 2**. Line 117, A is not defined.
>
> **Response:**
>
> - In Lemma 1, $A_\ell$ denotes the step size used in the gradient descent update at layer $\ell$.
> - We have clarified this lemma in Lines 117-118 of the revised paper.
>
>
> ---
> > **Question 3**. Showing the existence of such PVK matrices does not imply that your model would learn the same matrices. You would need to argue that your learning procedure would converge to desirable PVK matrices.
>
> **Response:**
>
> - We thank the reviewer for highlighting this point. Our existence proofs demonstrate that attention mechanisms in LLMs are expressive enough to realize PVK matrices that simulate gradient-based updates.
> - We would like to note that
>   - Recent theoretical work[5,6] has proven **that attention weights in pretrained Transformers do converge to such optimization-relevant matrices during in-context learning**, thereby providing theoretical support for our approach.
>   - Therefore, after establishing the existence of suitable PVK matrices, the ICL procedure can indeed converge to such desirable matrices in practice.
> - We have clarified this part in Appendix D (Lines 561-563) and Appendix F (Lines 617-619) of our revised paper.
>
> ----
> [1] Ariel Flint Ashery *et al.* Emergent social conventions and collective bias in LLM populations. Sci. Adv.11,eadu9368(2025).
>
> [2] Gao, Chen, et al. "Large language models empowered agent-based modeling and simulation: A survey and perspectives." *Humanities and Social Sciences Communications* 11.1 (2024): 1-24.
>
> [3] De Curtò, J., and I. De Zarzà. "LLM-Driven Social Influence for Cooperative Behavior in Multi-Agent Systems." *IEEE Access* (2025).
>
> [4] Ziems, Caleb, et al. "Can large language models transform computational social science?." *Computational Linguistics* 50.1 (2024): 237-291.
>
> [5] Ahn, Kwangjun, et al. "Transformers learn to implement preconditioned gradient descent for in-context learning." *Advances in Neural Information Processing Systems* 36 (2023): 45614-45650.
>
> [6] Gatmiry, Khashayar, et al. "Can looped transformers learn to implement multi-step gradient descent for in-context learning?" *arXiv preprint arXiv:2410.08292* (2024).

---

> > ### Comment · Reviewer_Jak5 · 2025-08-04
> >
> > Thank you for addressing my concerns. I believe this is an interesting work connecting ICL and SC, and I retain my positive evaluation. I do not have any further questions at this point.

---

> > > ### Author Response · Authors · 2025-08-04
> > >
> > > We sincerely thank you for your encouraging feedback.
> > >
> > > In the revised version of the paper, we will incorporate your suggested clarifications to further improve the presentation and explanation of our method.

---

### Note · Authors · 2025-08-11

Dear Reviewers, ACs, and SACs:

We sincerely appreciate the considerable time and effort you have dedicated in evaluating our work.

After reviewer-author discussion, we are encouraged that **all four reviewers are on the positive side of acceptance.** To facilitate the upcoming AC-Reviewer Discussions, we summarize the up-to-date status as follows.

- **Reviewer Jak5** praised our work as ```an interesting work connecting ICL and SC```.
  - After discussion and clarification, the reviewer **decided to** ```retain the positive evaluation with no further questions```.

- **Reviewer ZRpu** acknowledged the contributions of our work and *“believe the paper would make a good contribution”*.
  - During the discussion, **the reviewer expressed satisfaction with our responses and actively engaged with us**, offering constructive suggestions.
  - The reviewer **concluded** ```will update my rating and recommend an accept```.

- **Reviewer MEur** recognized *our contribution in connecting ICL with SC tasks* and highlighted ```our concise and effective introductions``` to the frameworks.
  - During the discussion, the reviewer provided detailed suggestions, such as adding specific prompts for each SC step, which we have implemented in the revision.
  - The reviewer **is satisfied with our responses and noted that** ```reading the paper and appreciating the actual methodological contribution much easier```.
  - Finally, reviewer **decided to** ```revise the recommendation to positive```.

- **Reviewer fu1o** provided a **positive** evaluation, recognizing our work as *interesting* and *useful* and **commenting** ```The writing is relatively clear and most theoretical results make sense to me.```
  - During the rebuttal process, the reviewer suggested exploring links to more complex strategic behaviors. We provided theoretical analysis and additional experiments to address this concern.

We are confident that our responses can thoroughly address the reviewers' concerns. We have carefully **revised the manuscript following each reviewer’s suggestions**, addressing the points one by one.

Thank you so much for the great efforts and thoughtfulness — we truly appreciate it!

Best,

The Authors of #7365

---

### Decision · Program_Chairs · 2025-09-17

**Decision:**

Accept (poster)

**Comment:**

The paper considers using LLMs for strategic classification. The authors propose a gradient free method based on in-context learning.
A theoretical analysis shows that both stages of strategic classification (inner optimization and outer optimization) can be realized through ICL.
The empirical evaluation shows that ICL-induced updates converge to gradient descent solutions. The proposed method also achieves superior classification accuracy compared to some traditional baselines.

The reviewers considered the connection between language models to strategic classification as interesting. The theoretical work, while limited, is also valuable. The empirical evaluation was also appreciated, especially the use of up-to-date LLMs.

There were several concern raised by the reviewers, but they were satisfied by the remedies provided in the rebuttal and the discussion. I feel that the paper can be accepted for NeurIPS, but I urge the authors to include the revisions suggested, including the additional experiments, and code. It is also important to make clear the limitations of their approach.